# Modeling Temporal scRNA-seq Data with Latent Gaussian Process and Optimal Transport

**Mehmet Yigit Balik** [1]   **Harri Lähdesmäki** [1]

## Abstract

Single-cell RNA sequencing provides insights into gene expression at single-cell resolution, yet inferring temporal processes from these static snapshot measurements remains a fundamental challenge. Current approaches utilizing neural differential equations and flows are sensitive to overfitting and lack careful considerations of biological variability. In this work, we propose a generative framework that models population trends using a latent heteroscedastic Gaussian process (GP) approximated by Hilbert space methods. To address the absence of genuine cell trajectories, we leverage an optimal transport (OT) objective that aligns generated and observed population distributions. Our method explicitly captures biological heterogeneity by incorporating cell-specific latent time and cell type conditioning to disentangle temporal asynchrony and trajectories to different cell types. We demonstrate state-of-the-art performance on complex interpolation and extrapolation benchmarks and introduce a novel gradient-based strategy for inferring perturbation trajectories.

## 1. Introduction

Single-cell RNA sequencing (scRNA-seq) has revolutionized our ability to resolve cellular heterogeneity by providing a high-resolution view of the gene expression landscape. However, understanding biological systems requires capabilities to model temporal processes, such as differentiation and development. A fundamental limitation of scRNA-seq is its destructive nature: since cells are destroyed during an experiment, their temporal evolution cannot be directly observed. As a result, the data consist only of population-level snapshots at observation time points, making it impossible

[1]Department of Computer Science, Aalto University, Espoo, Finland. Correspondence to: Mehmet Yigit Balik <mehmet.balik@aalto.fi>.

*Proceedings of the 43rd International Conference on Machine Learning*, Seoul, South Korea. PMLR 306, 2026. Copyright 2026 by the author(s).

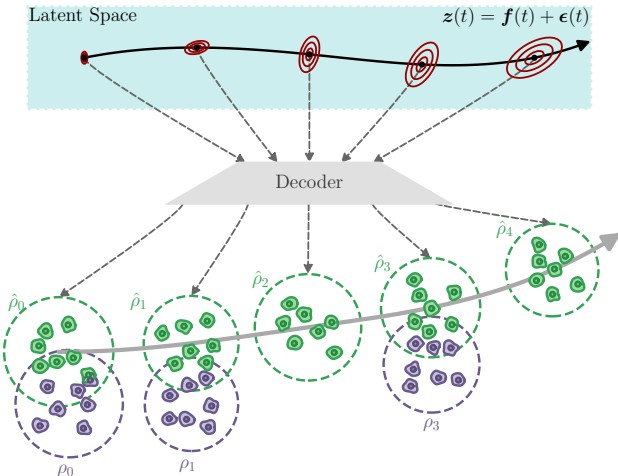

*Figure 1.* Overview of the proposed framework. A latent GP models smooth population-level temporal dynamics with heteroscedastic noise, which are decoded into distributions of cells at each time point and matched to observed data via OT.

to observe individual cells across time. This challenge motivates the development of computational methods that can reconstruct continuous underlying cellular dynamics from such sparse and discrete population-level observations.

While dynamical systems have been successfully employed to model biological processes, existing methods differ in how they represent the biological stochasticity. Deterministic frameworks, such as neural ordinary differential equations (ODEs) (Huguet et al., 2022; Zhang et al., 2024) and flow matching (Wang et al., 2025), effectively model trajectories but do not account for the variance inherent to biological processes. Conversely, stochastic differential equations (SDEs) offer an alternative for modeling biological stochasticity (Yeo et al., 2021; Jiang & Wan, 2024), though they necessitate a careful design of the diffusion to maintain stability (Oh et al., 2024). This requirement often restricts existing models to rigid assumptions. These considerations can lead to modeling choices that make it harder to capture time-varying uncertainty and to accommodate temporal asynchrony or multi-lineage structure within a single coherent and biologically plausible generative framework.

In this work, we propose Latent Gaussian Process with Optimal Transport (LGP-OT), a deep generative model with a heteroscedastic GP prior in the latent space to robustly capture global temporal trends, as illustrated in Figure 1. Similar to previous work (Huguet et al., 2022; Zhang et al., 2024), our framework leverages an OT loss that matches the distribution of generated cells and the observed cells, thus ensuring that our generated population faithfully reproduces the geometry and statistics of the biological population across time. Our approach models underlying dynamics while explicitly accounting for noise inherent to single-cell data. Moreover, we model temporal asynchrony via cell-specific latent time and incorporate categorical covariates into the kernel to account for multiple lineages. This enables learning of cell type-conditioned trajectories within a unified framework, which is not feasible with standard approaches. We use Hilbert space GP approximation (Solin & Särkkä, 2020), which scales linearly for large-scale, high-dimensional datasets (Balık et al., 2025).

Finally, generative models are often valued for their ability to simulate arbitrary conditioning, or hypothetical perturbations. While ODE-based models often utilize recursive numerical simulation, we extend the gradient-based latent variable perturbation strategy of Bjerregaard et al. (2025) to the temporal domain by optimizing entire latent trajectories to satisfy gene expression constraints. This enables inferring perturbation effects on temporal processes when recursive simulation is not available for perturbation analysis.

Our contributions are summarized as follows:

- We introduce LGP-OT with a scalable latent GP prior that incorporates cell-specific latent time and cell type-informed trajectories to capture complex biological heterogeneity, achieving state-of-the-art performance on interpolation and extrapolation tasks on real data.

- We employ an OT objective to learn GP-based population dynamics from a time series of scRNA-seq data.

- We extend gradient-based perturbation to GP-based temporal models to infer perturbation effects.

## 2. Related Work

**Variational autoencoders (VAEs) for scRNA-seq.** VAEs (Kingma & Welling, 2014) have become instrumental in the analysis of high-dimensional single-cell transcriptomics. Popular tools include, e.g. scVI (Lopez et al., 2018) and scVAE (Grønbech et al., 2020) that are used to learn low-dimensional latent representations that capture the underlying biological variability while effectively handling the sparsity, nuisance factors, and noise inherent in scRNA-seq data. However, capturing dynamic processes requires extending these methods to account for temporal dependencies.

**GP prior VAEs.** Combining the flexibility of GPs with the representational power of deep generative models has been a fruitful direction for time series analysis. GP prior VAEs (Casale et al., 2018) have been extensively developed to model time series data (Fortuin et al., 2020; Jazbec et al., 2021; Ramchandran et al., 2021; Tran et al., 2023; Balık et al., 2025; Shi et al., 2025), where the latent correlation structure is explicitly defined across time points and other covariates. While these methods excel at handling irregularly sampled high-dimensional time series, they assume that the samples are linked across time (i.e., tracking the same object). However, cells are destroyed in scRNA-seq experiments, yielding only a snapshot of the population at each time point. Consequently, standard GP prior VAE models that rely on direct sample-to-sample correlations are not directly applicable to the unpaired nature of single-cell data.

**Learning scRNA-seq dynamics.** Several approaches have been proposed to infer temporal dynamics from static snapshots. GPfates (Lönnberg et al., 2017) reconstructs differentiation trajectories by projecting single-cell data into a low-dimensional space using a GPLVM, infers a continuous pseudotime variable to order cells along their developmental progression, and applies an overlapping mixture of GPs as a function of this pseudotime to model branching events. GPfates can probabilistically assign cells to specific lineages and trace dynamic gene expression changes across diverging fates. Other methods focus on predicting gene expression at unobserved timepoints. Waddington-OT (Schiebinger et al., 2019) utilizes OT to predict cell-cell couplings, while TrajectoryNet (Tong et al., 2020) employs continuous normalizing flows to model paths between cell populations. However, both are limited to interpolation between observed endpoints and cannot extrapolate beyond the measured time range. To address extrapolation, PRESCIENT (Yeo et al., 2021) combines principal component analysis (PCA) with neural SDEs to model differentiation, though PCA-based representations may fail to capture complex cellular heterogeneity. Similarly, MIOFlow (Huguet et al., 2022) integrates a geodesic VAE with neural ODEs to retain cellular variations, but it fixes the encode-decode mappings prior to learning dynamics, which may limit its ability to generalize to unobserved time points with shifting distributions. scNODE (Zhang et al., 2024) integrates VAEs with neural ODEs and employs a dynamic regularization term to ensure the latent space is robust to distribution shifts, thereby improving extrapolation to unobserved time points. PI-SDE (Jiang & Wan, 2024) extends these dynamic frameworks by incorporating the least action principle into a physics-informed neural SDE model, allowing it to reconstruct an interpretable potential energy landscape governing cellular differentiation. Concurrently, simulation-free paradigms based on flow matching (Lipman et al., 2023; Tong et al., 2024a;b) have emerged to mitigate the computational burden of numeri-

cal integration in high dimensions by directly constructing probability paths. Furthermore, addressing varying sizes of cell populations, where proliferation and apoptosis violate mass conservation, unbalancedness-aware methods (Pariset et al., 2023; Sha et al., 2024; Zhang et al., 2025) have been developed to explicitly decouple mass variation from the state transport. Most recently, VGFM (Wang et al., 2025) unifies these concepts by jointly learning velocity and growth via flow matching, offering a scalable framework rooted in semi-relaxed OT for unbalanced populations.

# 3. Preliminaries

**Problem formulation and notation.** We consider a data distribution $\rho_t$ at each time point $t \in \mathcal{T} \subset \mathbb{R}$. At time $t$ we observe $n_t$ cells $\{\boldsymbol{x}_{t,i} \sim \rho_t\}_{i=1}^{n_t}$, where $\boldsymbol{x} \in \mathcal{X} = \mathbb{R}^G$ and $G$ is the number of genes. These samples form the gene expression matrix $\boldsymbol{X}_t \in \mathbb{R}^{n_t \times G}$ at time $t$, with the total number of observed cells being $N = \sum_{t \in \mathcal{T}} n_t$.

We are interested in modeling temporal dynamics in $L$-dimensional latent space $\mathcal{Z} = \mathbb{R}^L$ with $L \ll G$. The objective is to learn a generative model with predicted distribution $\hat{\rho}_t$ by matching that with the underlying distributions $\rho_t$ for each $t$. To this end, we generate $\hat{n}_t$ cells $\{\hat{\boldsymbol{x}}_{t,i} \sim \hat{\rho}_t\}_{i=1}^{\hat{n}_t}$, where $\hat{\boldsymbol{x}} \in \mathcal{X}$, forming the predicted gene expression matrix $\hat{\boldsymbol{X}}_t \in \mathbb{R}^{\hat{n}_t \times G}$. These generated cells originate from latent variables at time $t$, denoted as $\boldsymbol{Z}_t \in \mathbb{R}^{\hat{n}_t \times L}$.

## 3.1. Optimal Transport

OT provides a powerful and geometrically-grounded framework for comparing two probability distributions, namely $\rho$ and $\hat{\rho}$ (Villani et al., 2008). The core idea is to find the minimal cost to transport the mass from one probability distribution to another. A common choice for the cost function, $c : \mathcal{X} \times \mathcal{X} \to \mathbb{R}_+$, is the $p^{\text{th}}$ power of the Euclidean distance, $c(\boldsymbol{x}, \hat{\boldsymbol{x}}) = d(\boldsymbol{x}, \hat{\boldsymbol{x}})^p = \|\boldsymbol{x} - \hat{\boldsymbol{x}}\|_2^p$. This cost function reduces the Kantorovich formulation of the OT distance between $\rho$ and $\hat{\rho}$ to the $p$-Wasserstein distance

$$W_p^p(\rho, \hat{\rho}) \triangleq \inf_{\pi \in \Pi(\rho, \hat{\rho})} \int_{\mathcal{X}^2} d(\boldsymbol{x}, \hat{\boldsymbol{x}})^p \, \mathrm{d}\pi,$$

where $\Pi(\rho, \hat{\rho})$ is the set of all joint probability distributions (i.e. transport plans) $\pi$ on $\mathcal{X}^2$ with marginals $\rho$ and $\hat{\rho}$.

Calculating the Wasserstein distance using only samples from distributions becomes a discrete OT problem, which can be formulated as a linear program. To overcome the cubical scaling of the discrete OT problem (Peyré et al., 2019), it is common to add an entropic regularization to the OT objective (Cuturi, 2013), which makes the problem strongly convex and solvable in approximately linear time using the iterative Sinkhorn algorithm (Sinkhorn, 1964; Sinkhorn & Knopp, 1967). The resulting regularized OT objective is

defined as follows, noting that for observed snapshots, the integral reduces to a discrete evaluation on empirical samples drawn from population distributions at each time point:

$$W_{p,\gamma}^p(\rho, \hat{\rho}) \triangleq \inf_{\pi \in \Pi(\rho, \hat{\rho})} \int_{\mathcal{X}^2} d(\boldsymbol{x}, \hat{\boldsymbol{x}})^p \, \mathrm{d}\pi + \gamma \cdot \mathrm{KL}(\pi \| \rho \otimes \hat{\rho}). \tag{1}$$

## 3.2. Gaussian Processes

A GP is a stochastic process that defines a distribution over functions, typically as a function of time $t$. GP is fully specified by a mean function $\mu(t)$, which is generally considered as $0$ (we also use this convention for the rest of the paper) and a kernel function $k(t, t')$ (Rasmussen & Williams, 2006). The kernel encodes properties of the function, such as smoothness, by defining the covariance between function values at different inputs. By definition, for any finite set of inputs $\boldsymbol{t} = [t_1, \ldots, t_N]^\top$ the output values $f(\boldsymbol{t}) = [f(t_1), \ldots, f(t_N)]^\top$ have a joint Gaussian distribution $f(\boldsymbol{t}) \sim \mathcal{N}(\boldsymbol{0}, \boldsymbol{\Sigma})$, where $\boldsymbol{\Sigma}_{ij} = k(t_i, t_j)$ is $N \times N$ covariance matrix. We write $f(t) \sim \mathcal{GP}(\mu(t), k(t, t'))$.

## 3.3. Gaussian Process Prior Variational Autoencoders

To effectively model complex temporal high-dimensional data, the standard VAE framework can be extended by replacing the standard normal i.i.d. prior with a GP (Casale et al., 2018). Formally, we define a mapping from time $t \in \mathbb{R}$ to the latent space $\mathcal{Z}$, denoted $\boldsymbol{f} : \mathbb{R} \to \mathcal{Z}$, where $\boldsymbol{f}(t) = [f_1(t), \ldots, f_L(t)]^\top$. This function is governed by a multi-output GP prior, $\boldsymbol{f}(t) \sim \mathcal{GP}(\boldsymbol{0}, \boldsymbol{K}(t, t'))$, where $\boldsymbol{K}(t, t')$ is cross-covariance function. GP prior VAE models assume that this GP prior factorizes across the $L$ latent dimensions (Casale et al., 2018; Fortuin et al., 2020)

$$p(\boldsymbol{f}(\boldsymbol{t})) = \prod_{l=1}^{L} \mathcal{N}(\boldsymbol{0}, \boldsymbol{\Sigma}_l), \tag{2}$$

where $\boldsymbol{f}(\boldsymbol{t}) = [f_1(\boldsymbol{t}), \ldots, f_L(\boldsymbol{t})]^\top \in \mathbb{R}^{L \times N}$, and $\boldsymbol{\Sigma}_l$ is the covariance matrix for the $l^{\text{th}}$ dimension, derived from kernel $k_l$. Compared to existing multi-output GP models, such as the linear model of coregionalization (Alvarez et al., 2012), this factorization assumption comes with no loss of generality because a neural network decoder has enough capacity to model dependencies between the latent dimensions.

## 3.4. Hilbert Space Methods for Gaussian Processes

Evaluating the exact GP prior scales as $\mathcal{O}(N^3)$, which necessitates approximation techniques. While sparse GP methods based on inducing points are commonly used (Titsias, 2009), our approach leverages a scalable approximation rooted in Hilbert space methods (Solin & Särkkä, 2020). The central idea is to project the non-parametric GP onto a finite-dimensional subspace defined by a set of $M$ basis

functions $\boldsymbol{\phi}(t) = [\phi_1(t), \ldots, \phi_M(t)]^\top$. This procedure effectively recasts the GP into a tractable parametric form, which inherits a principled Bayesian prior over its weights.

Specifically, for a stationary kernel $k(t, t')$, the corresponding GP prior for function $f(t)$ is approximated as a linear combination of these Hilbert space basis functions

$$f(t) \approx \tilde{f}(t) = \boldsymbol{a}^\top \boldsymbol{\phi}(t) = \sum_{m=1}^{M} a_m \phi_m(t). \qquad (3)$$

We summarize the benefits of this approach below and provide full details in Appendix A (see also (Solin & Särkkä, 2020)). Basis functions are derived from the eigendecomposition of the Laplace operator. Consequently, the GP prior placed on the function $f$ is translated into a multivariate Gaussian prior on the $M$ random weights $\boldsymbol{a} \in \mathbb{R}^M$, expressed as $\boldsymbol{a} \sim \mathcal{N}(\boldsymbol{0}, \boldsymbol{S})$. This formulation proves particularly advantageous as the prior covariance matrix $\boldsymbol{S}$ is diagonal. The entries $[\boldsymbol{S}]_{mm}$ are specified by the kernel's spectral density $s(\cdot)$, the Fourier transform of $k$, evaluated at the square root of the Laplacian eigenvalues $\sqrt{\lambda_m}$ corresponding to each basis function $\phi_m$. This provides an elegant and efficient link, allowing the original kernel's hyperparameters (such as lengthscale and variance) to directly parameterize the weight prior $p(\boldsymbol{a})$. We can compactly denote this diagonal covariance matrix, which depends on the hyperparameters and eigenvalues $\boldsymbol{\lambda} = \{\lambda_m\}_{m=1}^{M}$, as $\boldsymbol{S}(\sqrt{\boldsymbol{\lambda}})$. All $\boldsymbol{\phi}(t)$, $\boldsymbol{\lambda}$ and $s(\cdot)$ have closed-form expressions.

# 4. Methods

## 4.1. Latent Hilbert Space Gaussian Process Prior

In LGP-OT, we model the temporal dynamics of high-dimensional gene expression at the population level using a multi-output GP in a lower-dimensional latent space, where we denote the latent variables as a function of time, $\boldsymbol{z}(t) \in \mathcal{Z}$. Since the level of biological variability in a temporal process is unlikely to be constant (e.g., it may increase during critical developmental state transitions), we adopt a heteroscedastic model, where the variability at the population level can change over time. We model this latent trajectory for a latent dimension $l$ as

$$z_l(t) = f_l(t) + \epsilon_l(t), \quad \epsilon_l(t) \sim \mathcal{N}(0, \varsigma_l(t)^2),$$

and assume a GP prior for the underlying signal $f_l(t)$ as well as for the log of the zero-mean noise standard deviation $\log(\varsigma_l(t))$ (to ensure positivity). We provide equations for $f_l(t)$ for the remainder of this section, yet all provided equations are analogous for $\log(\varsigma_l(t))$.

We resort to the Hilbert space methods for GP for scalability. While our framework is compatible with any stationary kernel amenable to Hilbert space approximation (Solin &

Särkkä, 2020), we assume squared exponential (SE) covariance function $k_{\ell_l, \sigma_l}(t, t') = \sigma_l^2 \exp(-\frac{\|t - t'\|^2}{2\ell_l^2})$ which has two hyperparameters: lengthscale $\ell_l$, and signal variance $\sigma_l^2$. Regarding the priors for the hyperparameters, we assume that the time points are standardized to unit scale and assign Lognormal$(0, 1)$ priors to both the $\ell_l$ (Timonen et al., 2021) and $\sigma_l$ (Balık et al., 2025) for non-negativity.

The signal function is represented as in Equation (3) with $M_f$ basis functions $\boldsymbol{\phi}_f(t)$

$$\tilde{f}_l(t) = \boldsymbol{a}_{f,l}^\top \boldsymbol{\phi}_f(t), \quad \boldsymbol{a}_{f,l} \sim \mathcal{N}\left(\boldsymbol{0}, \boldsymbol{S}_{\ell_{f,l}, \sigma_{f,l}}\left(\sqrt{\boldsymbol{\lambda}_f}\right)\right).$$

Adhering to the factorized prior in Equation (2), the Hilbert space approximation for the $L$-dimensional latent space is obtained by applying the basis function approximation independently to each of the latent dimensions (assuming the same number of basis functions per latent dimension)

$$\tilde{\boldsymbol{f}}(t) = \boldsymbol{A}_f \boldsymbol{\phi}_f(t), \qquad (4)$$

where $\boldsymbol{A}_f \in \mathbb{R}^{L \times M_f}$ is the weight matrix for the $L$ latent dimensions. The factorized prior in Equation (2) also implies to the prior of $\boldsymbol{A}_f$ that is given in Equation (5) below. This formulation provides a computationally efficient, heteroscedastic GP model for the latent trajectories, enabling us to simultaneously learn the smooth trajectories and their time-varying biological variability.

## 4.2. Generative Process and Optimization Objective

We denote the set of all global LGP-OT parameters as $\Phi = \{\boldsymbol{A}_f, \boldsymbol{A}_\varsigma, \boldsymbol{\ell}_f, \boldsymbol{\ell}_\varsigma, \boldsymbol{\sigma}_f, \boldsymbol{\sigma}_\varsigma\}$, which fully defines the latent temporal functions. Letting $j \in \{f, \varsigma\}$, the generative process for a predicted cell $i$ at time $t$ is defined as follows

$$\boldsymbol{\ell}_j, \boldsymbol{\sigma}_j \sim \prod_{l=1}^{L} \text{Lognormal}(0, 1),$$

$$\boldsymbol{A}_j \mid \boldsymbol{\ell}_j, \boldsymbol{\sigma}_j \sim \prod_{l=1}^{L} \mathcal{N}\left(\boldsymbol{a}_{j,l} \mid \boldsymbol{0}, \boldsymbol{S}_{\ell_{j,l}, \sigma_{j,l}}\left(\sqrt{\boldsymbol{\lambda}_j}\right)\right), \quad (5)$$

$$\boldsymbol{\xi}_{i,t} \sim \mathcal{N}(\boldsymbol{0}, \boldsymbol{I}_L),$$

$$\boldsymbol{z}_i(t; \Phi) = \boldsymbol{A}_f \boldsymbol{\phi}_f(t) + \exp(\boldsymbol{A}_\varsigma \boldsymbol{\phi}_\varsigma(t)) \odot \boldsymbol{\xi}_{i,t}, \quad (6)$$

$$\hat{\boldsymbol{x}}_{t,i} = h_\theta(\boldsymbol{z}_i(t; \Phi)),$$

where exponentiation $(\exp)$ and product $(\odot)$ are element-wise, and $h_\theta : \mathcal{Z} \to \mathcal{X}$ is a neural network decoder that maps latent states to the high-dimensional gene expression space. Unlike VAEs, which define an explicit likelihood model in the $\mathcal{X}$ domain, the function $h_\theta$ operates as a push-forward mapping. The predicted distribution $\hat{\rho}_t$ is therefore the distribution provided by $h_\theta(\boldsymbol{z}_i(t; \Phi))$.

Our objective function combines the geometric, OT-based population matching with variational regularization. We

utilize the entropy-regularized discrete Wasserstein distance $W_{p,\gamma}^p(\rho_t, \hat{\rho}_t)$ to match the observed samples from $\rho_t$ and generated samples from $\hat{\rho}_t$. Simultaneously, we learn a variational distribution of $\Phi$ by regularizing the discrete OT objective with an additional KL divergence between the variational distribution and the prior. To achieve this joint optimization, we adopt a variational framework (Hoffman et al., 2013). We use a mean-field variational distribution $q_\Psi(\Phi)$, where we use Gaussians for $\boldsymbol{A}_f, \boldsymbol{A}_\varsigma$ and log-normal for the other parameters. Our objective function consists of the *expectation* of the OT cost w.r.t. $q_\Psi(\Phi)$ and the KL divergence of variational distribution $q_\Psi(\Phi)$ from the prior $p(\Phi)$

$$\mathcal{L} = \underbrace{\mathbb{E}_{q_\Psi(\Phi)}\left[\sum_{t \in \mathcal{T}} W_{p,\gamma}^p(\rho_t, \hat{\rho}_t)\right]}_{\text{Expected transport cost}} + \underbrace{\mathrm{KL}(q_\Psi(\Phi)\|p(\Phi))}_{\text{Regularization}}. \quad (7)$$

The predicted distribution $\hat{\rho}_t$ in the expected transport cost term depends on the latent temporal functions sampled from the variational distribution, $\Phi \sim q_\Psi(\Phi)$, and the parameters of the push-forward mapping $\theta$. The regularization term ensures that the learned temporal dynamics adhere to the smoothness and structural constraints imposed by the GP priors. Note that prior $p(\Phi)$ has a specific hierarchical structure because the priors for $\boldsymbol{A}_j$, $j \in \{f, \varsigma\}$, are conditional on hyperparameters $\boldsymbol{\ell}_j, \boldsymbol{\sigma}_j$ via the spectral density of the kernel $k$. Details of computing the regularization term are given in Appendix B. Overall, this formulation is analogous to the Wasserstein autoencoder (WAE) framework (Tolstikhin et al., 2018), which establishes generative modeling in terms of minimizing the $W_2(\rho, \hat{\rho})$. While WAE typically relaxes the $W_2$ distance into pairwise reconstruction costs with a latent penalty that matches the marginal distribution of latent representations with the prior distribution, we employ the Wasserstein distance directly in the observation space to overcome the lack of pairwise correspondence, while retaining the KL divergence to regularize the latent temporal functions toward the GP prior.

The full objective is optimized jointly with respect to the generator parameters $\theta$ and the variational parameters $\Psi$. In practice, we estimate expectation $\mathbb{E}_{q_\Psi(\Phi)}[\cdot]$ using Monte Carlo sampling. We employ the reparameterization trick (Kingma & Welling, 2014) for the variational distribution $q_\Psi$, enabling gradient-based optimization of $\Psi$. The regularized distance $W_{p,\gamma}^p$ is computed between samples from $\rho_t$ (i.e., minibatches of data $\boldsymbol{X}_t$) and $\hat{\rho}_t$ (i.e., generated samples $\hat{\boldsymbol{X}}_t$). This makes the entire objective differentiable and solvable using stochastic gradient descent. We provide the mathematical details in Appendix B.

**Decoder-only design.** LGP-OT intentionally omits an encoder. Because scRNA-seq is destructive, the movement of any single cell through gene expression space over time is not observed. Gene expression information from observed cells enters the model through the OT objective, where gradients of the Wasserstein distance with respect to the variational parameters propagate the observed population structure back into the latent trajectory. Encoder-based approaches couple two roles into the encoder: learning a low-dimensional representation and providing initial conditions for the dynamical system. However, an encoder trained on observed time points may not generalize to unobserved times with shifted distributions (Zhang et al., 2024). Our framework avoids this issue by construction, as the decoder only needs to map from a smoothly varying latent space whose structure is governed by the GP prior. Per-cell embeddings can still be recovered if needed via non-amortized inference, by inverting the push-forward mapping for individual cells at any time point (see also Section 4.5).

### 4.3. Modeling Cell-Specific Time Variability

To account for the inherent temporal asynchrony across individual cells, we propose to estimate cell-specific time $\tau_n \in \mathbb{R}$ for each cell $n$. Following previous work (Reid & Wernisch, 2016; Lönnberg et al., 2017), the measurement time $t_n$ is treated as an informative prior mean for $\tau_n$

$$p(\tau_n|t_n) = \mathcal{N}(\tau_n|t_n, \delta^2),$$

where $\delta$ is a fixed prior standard deviation. We then infer this latent time by treating $\boldsymbol{\tau} = [\tau_1, \ldots, \tau_N]$ as an additional set of latent variables included in $\Phi$ and extending $\Psi$ to include their corresponding variational parameters. We introduce a factorized variational distribution $q_\Psi(\boldsymbol{\tau}) = \prod_{n=1}^N q_\Psi(\tau_n)$, where each $q_\Psi(\tau_n)$ is a Gaussian

$$q_\Psi(\tau_n) = \mathcal{N}(\tau_n|\mu_n, \nu_n^2).$$

In the generative process, the latent state $\boldsymbol{z}_n$ is now generated by first sampling the cell-specific time $\tau_n$ and then evaluating the functions in Equation (6). Incorporating the variational distributions $q_\Psi(\boldsymbol{\tau})$ and their priors $p(\boldsymbol{\tau})$ into the optimization objective in Equation (7) modifies the first term to include an expectation also over $q_\Psi(\boldsymbol{\tau})$ and adds a new KL divergence term, $\sum_{n=1}^N \mathrm{KL}(q_\Psi(\tau_n)\|p(\tau_n|t_n))$, to the objective's regularization component.

**Distinction from pseudotime.** Classical pseudotime methods infer an ordering of cells along a developmental axis, typically from a single snapshot where real temporal annotations are absent. In contrast, our cell-specific latent time operates in a time-series setting where each $\tau_n$ is anchored to the recorded measurement time $t_n$ via the informative prior, capturing small deviations that reflect within-snapshot temporal asynchrony rather than constructing an ordering. This reflects the biological reality that cells within the same snapshot may differ slightly, while remaining grounded in the experimentally recorded time.

## 4.4. Cell Type Conditioning

While our primary focus has been on continuous temporal dynamics, biological data often involves categorical covariates $c \in \mathcal{C} = \{1, \ldots, C\}$, such as cell types. We can utilize GPs to model those covariates (Qian et al., 2008) as well. It is possible to represent this GP as a linear model similar to Equation (4) by utilizing eigendecomposition of the kernel matrix $\boldsymbol{K}_\mathcal{C} \in \mathbb{R}^{C \times C}$. Specifically, as $\boldsymbol{K}_\mathcal{C}$ is symmetric, it admits the spectral decomposition $\boldsymbol{K}_\mathcal{C} = \boldsymbol{U} \boldsymbol{D} \boldsymbol{U}^\top$, where $\boldsymbol{D} = \text{diag}(d_1, \ldots, d_C)$ contains the eigenvalues and $\boldsymbol{U}$ contains the eigenvectors. The categorical basis functions $\boldsymbol{u}(c) \in \mathbb{R}^C$ are defined as the rows of $\boldsymbol{U}$ corresponding to category $c$. To capture the interplay between temporal dynamics and categorical cell type covariates, we employ a product kernel structure $k((t, c), (t', c')) = k(t, t')[\boldsymbol{K}_\mathcal{C}]_{c, c'}$ (Saves et al., 2023; Timonen et al., 2021; Balık et al., 2025). Leveraging the property that the eigenbasis of a product kernel corresponds to the Kronecker product of the individual bases, we can express the joint function approximation as

$$\tilde{f}_l(t, c) = \boldsymbol{a}_{f,l}^\top (\boldsymbol{\phi}_f(t) \otimes \boldsymbol{u}_f(c)),$$

where $\otimes$ denotes the Kronecker product. Consequently, the prior over the weights $\boldsymbol{a}_{f,l} \in \mathbb{R}^{M \cdot C}$ is determined by the Kronecker product of the individual spectral densities

$$\boldsymbol{a}_{f,l} \sim \mathcal{N}\left(\boldsymbol{0}, \boldsymbol{S}_{\ell_{f,l}, \sigma_{f,l}}\left(\sqrt{\boldsymbol{\lambda}}\right) \otimes \boldsymbol{D}\right).$$

By utilizing the Kronecker product, we avoid constructing the full covariance matrix, maintaining the scalability benefits of the Hilbert space approximation while extending the model's expressivity to discrete biological factors.

## 4.5. Simulating Gene Perturbations via Gradients

The push-forward function $h_\theta$ learns a differentiable mapping from latent variables to gene expression, implicitly capturing the underlying manifold of the cellular states. Bjerregaard et al. (2025) simulate perturbations by shifting latent coordinates $\boldsymbol{z}(t)$ along the gradient of a target gene $g$: $\boldsymbol{z}_{i+1}(t) := \boldsymbol{z}_i(t) + \eta \cdot \nabla_{\boldsymbol{z}} h_\theta^{(g)}(\boldsymbol{z}_i(t))$, where $\eta$ scales the perturbation; negative and positive values simulate knockout and overexpression, respectively. However, this pointwise update ignores temporal dependencies. To address this, we extend the strategy to the temporal domain by modifying the base variational distribution $q_\Psi(\Phi)$ that governs the latent trajectories, thereby updating the entire function $\boldsymbol{z}(t; \Phi)$. We define $\rho_t^{(\boldsymbol{\alpha}, \mathcal{G})}$ to be the data distribution at a time $t$, where a set of genes $\mathcal{G}$ is perturbed with a vector of scaling factors $\boldsymbol{\alpha}$. We drive the trajectories towards these perturbed states by minimizing the expected OT cost between $\rho_t^{(\boldsymbol{\alpha}, \mathcal{G})}$ and $\hat{\rho}_t$. This yields the update rule: $\Psi_{i+1} := \Psi_i - \eta \cdot \nabla_{\Psi_i} \mathbb{E}_{q_{\Psi_i}(\Phi)} \left[ W_{p, \gamma}^p (\rho_t^{(\boldsymbol{\alpha}, \mathcal{G})}, \hat{\rho}_t) \right]$, where $\eta$ serves as the learning rate for the optimization.

## 5. Experiments

We utilize three publicly available scRNA-seq datasets covering various species and tissues to demonstrate the performance of our model: Zebrafish embryo (ZB) (Farrell et al., 2018), Drosophila (DR) (Calderon et al., 2022), and iPSC reprogramming in mice (SC) (Schiebinger et al., 2019).

The datasets vary in temporal resolution and size, presenting distinct challenges for modeling temporal dynamics. The ZB dataset consists of 38,731 cells across 12 time points, with cell types being available only at the last time point; the DR dataset contains 27,386 cells across 11 time points, along with cell types across all time points; and the SC dataset has 236,285 cells spanning 19 time points. The preprocessing pipeline is detailed in Appendix C.1.

To ensure a comprehensive analysis, we evaluate five different versions of our model, LGP-OT, including variants for latent cell-time modeling (Section 4.3), cell type-conditioned trajectory modeling (Section 4.4), and a combination of those two, if the datasets provide complete cell type annotations; additionally, we compare our methods against the homoscedastic version of our model and the following state-of-the-art baselines: **scNODE** (Zhang et al., 2024) integrates a VAE with neural ODEs via dynamic regularization. **MIOFlow** (Huguet et al., 2022) combines geodesic autoencoders with neural ODEs. **PRESCIENT** (Yeo et al., 2021) and **PI-SDE** (Jiang & Wan, 2024) utilize potential-driven and physics-informed SDEs, respectively. **VGFM** (Wang et al., 2025) uses flow matching based on semi-relaxed OT to jointly learn velocity fields and mass growth for unbalanced populations. LGP-OT and scNODE train a non-linear mapping between the high-dimensional observation space and latent space simultaneously with training the dynamics model in the latent space, while other baselines pre-train a linear (PCA) or non-linear dimension reduction, train dynamics in the fixed latent space, and transform model predictions back to the observation space using the pre-trained reverse mapping. We tuned all models to their optimal hyperparameters (Appendix C.2).

We evaluate the models' ability to predict gene expression distributions at unobserved time points. Following Zhang et al. (2024), we establish three distinct tasks of increasing difficulty for each dataset by withholding specific time points during training: (1) **Easy (interpolation):** Uniformly spaced time points within the measured range are removed to test the model's ability to recover intermediate states. (2) **Medium (extrapolation):** The final few time points are withheld to evaluate the model's capacity to forecast future cell population distributions beyond the training horizon. (3) **Hard (combined):** A combination of intermediate and final time points is removed, requiring the model to simultaneously interpolate and extrapolate under significant data sparsity. The specific hold-out time points for

*Table 1.* Performance comparison on the DR dataset across three tasks. Refer to Tables 7 and 8 for all datasets.

| Model | DR/Easy | | | DR/Medium | | |
|---|---|---|---|---|---|---|
| | $t = 4$ | $t = 6$ | $t = 8$ | $t = 8$ | $t = 9$ | $t = 10$ |
| LGP-OT | $26.35 \pm 0.06$ | $28.22 \pm 0.02$ | $30.74 \pm 0.03$ | $32.15 \pm 0.04$ | $32.52 \pm 0.08$ | $36.51 \pm 0.04$ |
| LGP-OT (w. type) | $26.46 \pm 0.08$ | $28.18 \pm 0.04$ | $\mathbf{30.59} \pm \mathbf{0.05}$ | $31.92 \pm 0.06$ | $31.79 \pm 0.10$ | $36.48 \pm 0.06$ |
| LGP-OT (w. time) | $26.25 \pm 0.13$ | $28.32 \pm 0.09$ | $30.82 \pm 0.05$ | $32.24 \pm 0.11$ | $\mathbf{31.54} \pm \mathbf{0.04}$ | $36.33 \pm 0.05$ |
| LGP-OT (w. time + type) | $\mathbf{26.09} \pm \mathbf{0.03}$ | $\mathbf{28.12} \pm \mathbf{0.04}$ | $30.59 \pm 0.06$ | $\mathbf{31.88} \pm \mathbf{0.07}$ | $31.63 \pm 0.09$ | $\mathbf{36.32} \pm \mathbf{0.20}$ |
| LGP-OT ($\varsigma(t) = 0.1$) | $26.85 \pm 0.19$ | $29.16 \pm 0.25$ | $31.90 \pm 0.28$ | $33.24 \pm 0.17$ | $33.05 \pm 0.20$ | $37.41 \pm 0.22$ |
| VGFM | $\mathbf{26.09} \pm \mathbf{0.19}$ | $28.34 \pm 0.07$ | $31.29 \pm 0.13$ | $33.73 \pm 0.04$ | $32.88 \pm 0.26$ | $37.06 \pm 0.47$ |
| scNODE | $26.58 \pm 0.12$ | $28.65 \pm 0.11$ | $31.46 \pm 0.15$ | $33.24 \pm 0.16$ | $33.67 \pm 0.38$ | $38.04 \pm 0.35$ |
| MIOFlow | $26.44 \pm 0.12$ | $28.55 \pm 0.06$ | $31.65 \pm 0.13$ | $33.60 \pm 0.12$ | $34.49 \pm 0.24$ | $38.76 \pm 0.82$ |
| PI-SDE | $26.70 \pm 0.03$ | $28.76 \pm 0.04$ | $31.39 \pm 0.07$ | $32.80 \pm 0.09$ | $33.49 \pm 0.37$ | $38.27 \pm 0.57$ |
| PRESCIENT | $30.16 \pm 0.09$ | $31.16 \pm 0.08$ | $32.48 \pm 0.07$ | $34.66 \pm 0.24$ | $33.26 \pm 0.22$ | $37.64 \pm 0.34$ |

| Model | DR/Hard | | | | | |
|---|---|---|---|---|---|---|
| | $t = 2$ | $t = 4$ | $t = 6$ | $t = 8$ | $t = 9$ | $t = 10$ |
| LGP-OT | $29.40 \pm 0.13$ | $30.61 \pm 0.07$ | $32.33 \pm 0.06$ | $33.78 \pm 0.02$ | $33.85 \pm 0.11$ | $38.17 \pm 0.20$ |
| LGP-OT (w. type) | $\mathbf{28.75} \pm \mathbf{0.04}$ | $\mathbf{30.39} \pm \mathbf{0.06}$ | $\mathbf{32.19} \pm \mathbf{0.01}$ | $\mathbf{33.75} \pm \mathbf{0.07}$ | $33.52 \pm 0.04$ | $37.82 \pm 0.23$ |
| LGP-OT (w. time) | $29.59 \pm 0.06$ | $30.48 \pm 0.06$ | $32.30 \pm 0.02$ | $33.83 \pm 0.11$ | $33.68 \pm 0.08$ | $\mathbf{37.71} \pm \mathbf{0.08}$ |
| LGP-OT (w. time + type) | $28.84 \pm 0.05$ | $30.48 \pm 0.07$ | $32.20 \pm 0.02$ | $33.77 \pm 0.08$ | $\mathbf{33.52} \pm \mathbf{0.07}$ | $37.88 \pm 0.30$ |
| LGP-OT ($\varsigma(t) = 0.1$) | $30.14 \pm 0.21$ | $31.28 \pm 0.21$ | $33.29 \pm 0.10$ | $34.93 \pm 0.09$ | $34.88 \pm 0.14$ | $38.79 \pm 0.33$ |
| VGFM | $30.27 \pm 0.05$ | $31.30 \pm 0.06$ | $33.36 \pm 0.03$ | $35.62 \pm 0.06$ | $35.36 \pm 0.20$ | $38.47 \pm 0.23$ |
| scNODE | $29.95 \pm 0.29$ | $30.71 \pm 0.16$ | $32.88 \pm 0.12$ | $34.82 \pm 0.23$ | $34.63 \pm 0.41$ | $38.26 \pm 0.51$ |
| MIOFlow | $29.74 \pm 0.18$ | $30.97 \pm 0.07$ | $32.84 \pm 0.08$ | $35.25 \pm 0.17$ | $36.78 \pm 0.34$ | $41.43 \pm 1.07$ |
| PI-SDE | $30.34 \pm 0.14$ | $30.82 \pm 0.09$ | $32.71 \pm 0.05$ | $34.18 \pm 0.17$ | $34.55 \pm 0.27$ | $38.90 \pm 0.54$ |
| PRESCIENT | $32.68 \pm 0.13$ | $32.36 \pm 0.11$ | $33.08 \pm 0.11$ | $35.51 \pm 0.21$ | $34.19 \pm 0.14$ | $38.12 \pm 0.15$ |

each task are provided in Table 3 (Appendix C.1). We evaluate predictions using the Wasserstein distance ($W_2$) against the entire held-out ground truth population. We report the mean and standard deviation across five iterations using 2000 generated samples from each model per test time point. The implementation of our model is provided at `https://github.com/YigitBalik/LGP-OT`.

### 5.1. Results

Table 1 presents results for the DR dataset, where similar trends were observed for other datasets (see Tables 7-8).

**Interpolation and extrapolation performance.** In the DR data, LGP-OT combines cell-time variability and cell type conditioning to achieve the lowest $W_2$ across time points (Table 1). Similarly, in the ZB dataset, our model outperforms the strongest baseline at all time points (Table 7). Cell-time variability modeling notably reduces $W_2$ distance during extrapolation. Finally, in the large-scale SC dataset, LGP-OT demonstrates robust generalization capabilities (Table 7). Incorporating heteroscedasticity provides consistent performance gains over the homoscedastic baseline, underscoring the necessity of modeling time-varying noise. Figure 20 visualizes the learned latent temporal func-

tions for the SC dataset, where the varying uncertainty bands across latent dimensions illustrate that the model learns non-constant noise structure rather than relying on a fixed noise assumption. Overall, across all three datasets, LGP-OT performs competitively in the easy and medium tasks.

**Modeling under sparsity.** This setting poses the most significant challenge, requiring models to reconstruct intermediate states while simultaneously forecasting future dynamics. LGP-OT maintains its robustness, consistently outperforming baselines, which tend to degrade in performance at later time points (Tables 1 and 8). In the DR hard task, incorporating cell types consistently surpasses the base model, suggesting that conditioning on cell types helps stabilize predictions when data is sparse (Table 1). Notably, in the SC dataset, the performance gap widens significantly at $t = 17$ and $t = 18$, where LGP-OT (w. time) achieves $W_2$ of 15.05 and 15.53, respectively, outperforming the most competitive baseline scNODE (17.26 and 18.14; Table 8). Models like PI-SDE and MIOFlow deteriorate to values exceeding 19.0. VGFM remains a competitive baseline across several time points, but its performance is less consistent and generally trails the strongest methods. Results show that LGP-OT effectively captures the underlying manifold even when substantial portions of the trajectory are withheld.

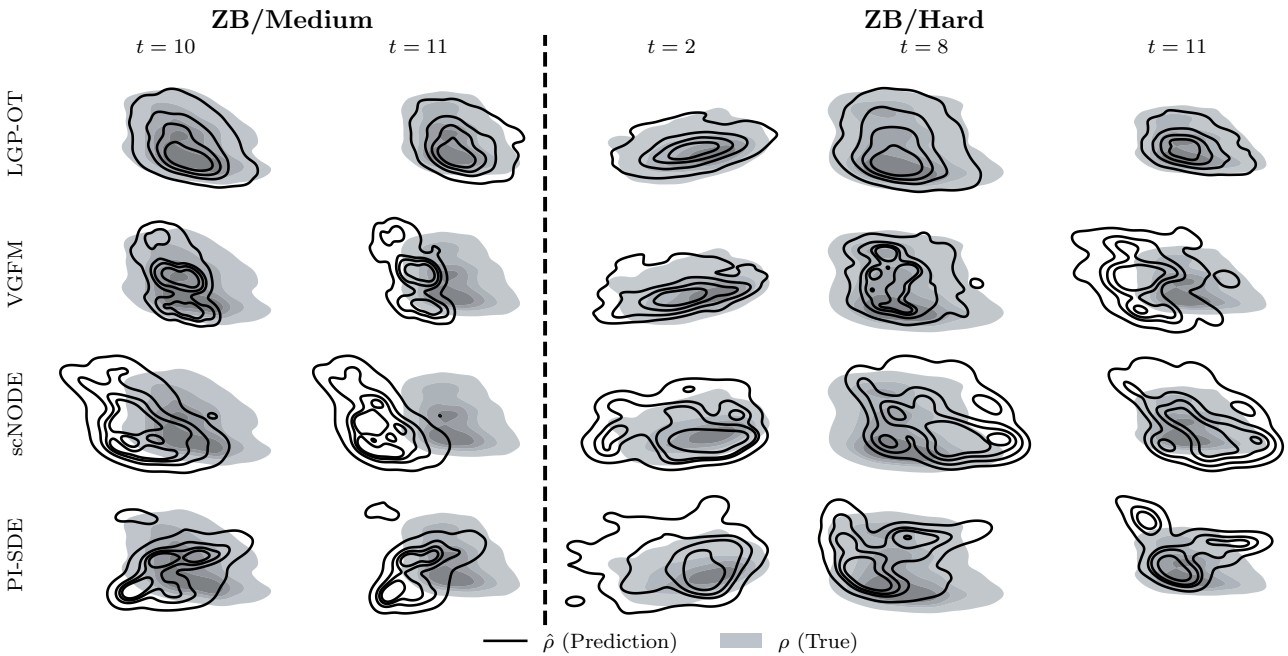

*Figure 2.* Visual comparison of predicted and observed cell distributions on the ZB dataset.

**Qualitative Analysis.** Figure 2 complements our quantitative results by visualizing predicted cell concentrations at selected time points for the ZB datasets (medium and hard tasks), where the gene expression space is projected onto the first two principal components and contours are generated using kernel density estimation. In the medium task, our model accurately captures the geometry of the ground truth. In contrast, baselines such as scNODE occasionally produce spatially misaligned contours, while PI-SDE tends to generate fragmented distributions. In the hard task, our method maintains structural coherence more effectively than competing approaches. As shown in Figure 2 (right), baselines exhibit distinct failure modes under sparsity: PI-SDE produces disjoint clusters that break the continuity, and VGFM yields distorted shapes. Detailed trajectory plots for all datasets and models for all time points are provided in Appendix C.5 (Figures 8-19). These visualizations consistently corroborate that our model preserves the underlying manifold better than baselines, even under significant sparsity.

**Perturbation analysis.** To validate the model's capacity to uncover key gene dependencies and investigate how the global temporal process adapts when genes are perturbed, we performed *in silico* perturbation experiments on the ZB dataset. We first trained LGP-OT on all time points and trained a multi-layer perceptron cell type classifier on the observed data at the terminal time point (evaluation of the classifier performance and detailed metrics provided in Appendix C.4). We then identify differentially expressed (DE)

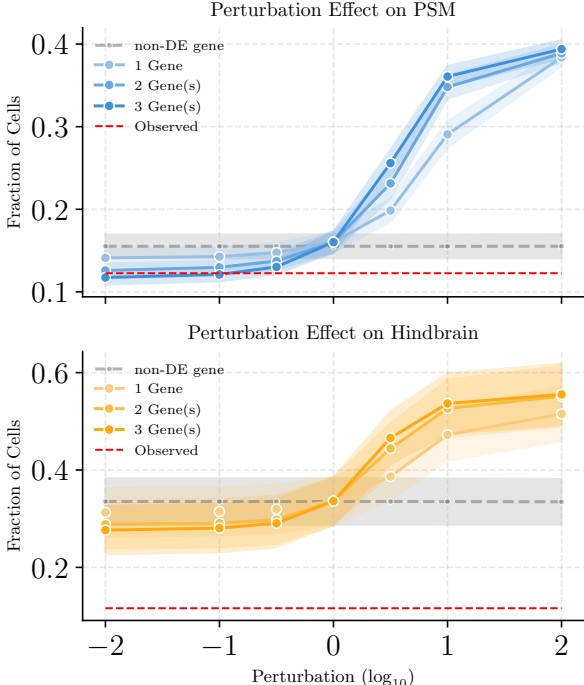

*Figure 3.* Effect of perturbing different numbers of DE genes of PSM and Hindbrain cells on the predicted fraction of cells classified into each type at the last time point of the ZB dataset. The dashed horizontal line marks the observed cell type ratio.

Inferred Trajectories under Gene Perturbations

Inferred Trajectories of Cell Types

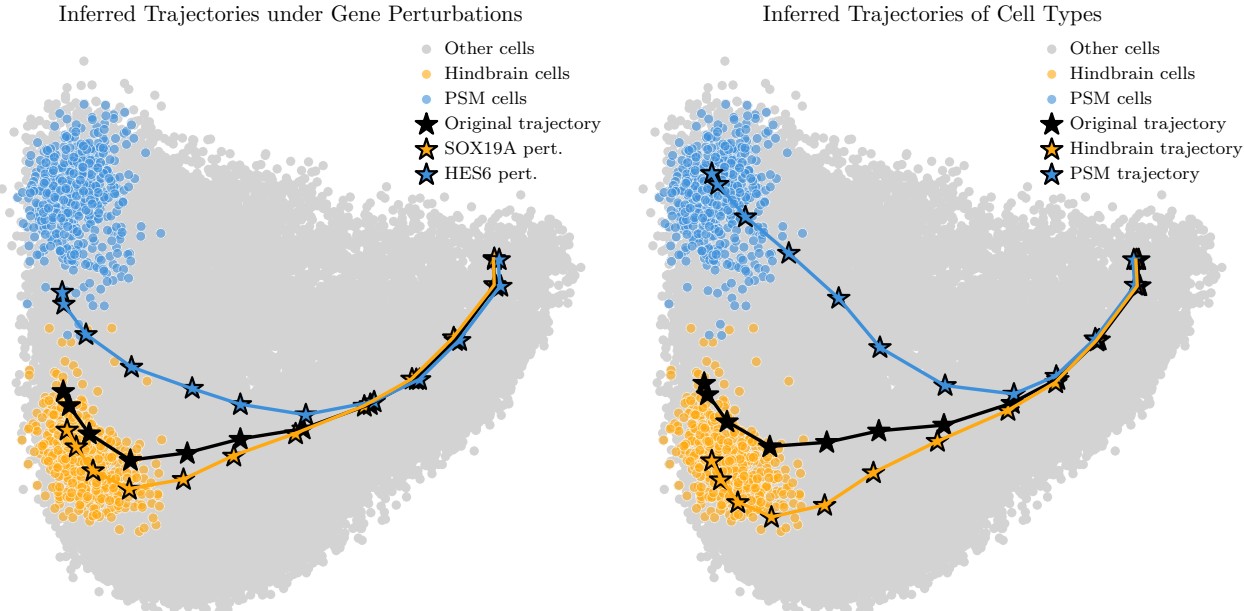

*Figure 4.* Inferred trajectories under gene and cell type-specific perturbations. Left: Inferred trajectories resulting from perturbing only the most DE gene of PSM and Hindbrain cells. The original model's mean trajectory steered toward the Hindbrain cluster by scaling up the *SOX19A* gene, or toward the PSM cluster by scaling up the *HES6* gene. Right: Inferred trajectories where the optimization target is defined by the distribution of the specific cell types themselves, rather than scaling specific genes.

genes of Presomitic Mesoderm (PSM) and Hindbrain cells at the terminal time point, where the top three DE genes are presented in Figure 6. We perturbed an increasing number of these genes with varying scaling factors ($\log_{10} \alpha$ from $-2$ to 2). As illustrated in Figure 3, increasing the scaling factor for lineage-specific drivers progressively biased the terminal cell distribution toward the targeted cell type. This effect intensified as more genes were perturbed, although the boost in target cell proportion saturated at the highest perturbation magnitude. In contrast, perturbing non-DE genes yielded no significant shift, as expected. Furthermore, visualizing the trajectory in Figure 4 (left) confirms that these perturbations effectively steer the global trajectory toward the expected lineage clusters. Complementing these gene-level perturbations, we also investigated cell type-specific guidance where the target distribution is defined by the distribution of specific cell types at the final time point (Figure 4, right). With the independent guidance for two cell types, the classifier identified $0.93 \pm_{0.01}$ of the generated 2000 cells as Hindbrain and $0.92 \pm_{0.02}$ as PSM. These results demonstrate that LGP-OT captures key gene dependencies, allowing it to predict changes in the trajectory of snapshots.

## 6. Conclusions

In this work, we presented a generative framework for reconstructing single-cell temporal dynamics by modeling global population trends via a scalable latent heteroscedastic

GP prior. By leveraging OT, our method effectively learns smooth, continuous trajectories from unpaired snapshots and robustly handles biological complexities such as temporal asynchrony and cell type-informed paths. Empirically, our approach establishes a new state-of-the-art on challenging interpolation and extrapolation tasks, outperforming existing neural ODE, SDE, and flow baselines. Furthermore, our gradient-based perturbation strategy for temporal models provides an alternative tool for perturbation analysis. The decoder-only design also addresses a structural limitation shared by many VAE-based methods, as it avoids encoder generalization failures at unobserved time points by construction. Our approach offers a significant and robust alternative for modeling temporal scRNA-seq data.

**Limitations and future work.** While LGP-OT achieves strong performance, several limitations point to promising future directions. Although cell type conditioning and cell-specific latent time partially address multi-lineage structure and temporal asynchrony, the model is not designed to explicitly capture branching events. Thus, integrating branching GPs into the latent space would be a natural extension. Moreover, the SE kernel assumes smooth, non-oscillatory dynamics: replacing it with a periodic kernel, which admits a similar low-rank approximation, would extend LGP-OT to cyclic biological processes. Finally, LGP-OT is limited to transcriptomic readouts. Extending it to multi-modal or spatial single-cell data remains an open problem.

## Acknowledgements

We would like to acknowledge the computational resources provided by the Aalto Science-IT. We would like to thank the anonymous reviewers for their thoughtful comments that strengthened this work. This work was supported by the Research Council of Finland (Flagship programme: Finnish Center for Artificial Intelligence FCAI and decision #359135).

## Impact Statement

Addressing the limitation of sparse and expensive temporal single-cell sequencing, this work introduces a generative framework that reconstructs continuous cellular dynamics from discrete population snapshots. The model enables the generation of synthetic gene expression profiles at any unobserved time point. We anticipate no foreseeable adverse societal effects.

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

# A. Hilbert Space Methods for Reduced-Rank Gaussian Process Regression

Here, we present the mathematical foundations of Hilbert space methods for reduced-rank GP regression introduced by Solin & Särkkä (2020). We consider only the unidimensional case where a stationary continuous kernel of a GP depends on a single continuous covariate, denoted as $t$, in line with our notation. A stationary covariance function can be written as a function of $r = t - t'$. In such a case, the covariance function can be represented by the spectral density of the kernel, which is the Fourier transform of the kernel according to Bochner's and Wiener-Khintchine theorems (Akhiezer & Glazman, 2013; Da Prato & Zabczyk, 2014; Rasmussen & Williams, 2006) as

$$k_{\ell,\sigma}(r) = \frac{1}{2\pi} \int_{-\infty}^{\infty} s_{\ell,\sigma}(\omega) e^{i\omega r} d\omega,$$

$$s_{\ell,\sigma}(\omega) = \int_{-\infty}^{\infty} k_{\ell,\sigma}(r) e^{-i\omega r} dr.$$

For the SE kernel used in this work, the spectral density has a closed-form expression given by:

$$k_{\ell,\sigma}(t, t') = \sigma^2 \exp\left(-\frac{\|t - t'\|^2}{2\ell^2}\right)$$

$$s_{\ell,\sigma}(\omega) = \sigma^2 \sqrt{2\pi}\ell \exp\left(-\frac{\ell^2\omega^2}{2}\right).$$

The core insight of the Hilbert space approximation is to interpret the spectral density $s_{\ell,\sigma}(\cdot)$ as a transfer function of a pseudo-differential operator associated with the covariance function. Specifically, for isotropic kernels, the spectral density depends only on the magnitude of the frequency $|\omega|$. Since the Fourier transform of the Laplace operator $\nabla^2$ is $-|\omega|^2$, the covariance operator can be approximated via the eigendecomposition of the Laplacian in a compact domain $\Omega \subset \mathbb{R}$ subject to boundary conditions.

We define the domain $\Omega = [-J, J]$ such that it contains the input domain, where $J$ is a boundary parameter chosen to be slightly larger than the data range to mitigate boundary effects (Riutort-Mayol et al., 2023). The approximation utilizes the eigenfunctions $\phi_m(t)$ and eigenvalues $\lambda_m$ of the negative Laplacian operator $-\nabla^2$ subject to Dirichlet boundary conditions:

$$\begin{cases} -\nabla^2 \phi_m(t) = \lambda_m \phi_m(t), & t \in \Omega \\ \phi_m(t) = 0, & t \in \partial\Omega. \end{cases}$$

The solutions to this eigenvalue problem in the unidimensional domain are known analytically:

$$\lambda_m = \left(\frac{\pi m}{2J}\right)^2, \quad \phi_m(t) = \frac{1}{\sqrt{J}} \sin\left(\sqrt{\lambda_m}(t + J)\right), \quad m = 1, 2, \ldots$$

By truncating the expansion at $M$ terms, the kernel $k_{\ell,\sigma}(t, t')$ is approximated as:

$$k_{\ell,\sigma}(t, t') \approx \sum_{m=1}^{M} s_{\ell,\sigma}(\sqrt{\lambda_m})\phi_m(t)\phi_m(t').$$

This series expansion allows us to approximate the GP $f(t) \sim \mathcal{GP}(0, k_{\ell,\sigma}(t, t'))$ as a finite-dimensional linear model:

$$f(t) \approx \sum_{m=1}^{M} a_m \phi_m(t) = \boldsymbol{a}^\top \boldsymbol{\phi}(t),$$

where the weights are independent Gaussian random variables $a_m \sim \mathcal{N}(0, s_{\ell,\sigma}(\sqrt{\lambda_m}))$ or, equivalently, $\boldsymbol{a} \sim \mathcal{N}\left(\boldsymbol{0}, \boldsymbol{S}_{\ell,\sigma}\left(\sqrt{\boldsymbol{\lambda}}\right)\right)$, where $\boldsymbol{S}_{\ell,\sigma}\left(\sqrt{\boldsymbol{\lambda}}\right) = \mathrm{diag}\left(s_{\ell,\sigma}(\sqrt{\lambda_1}), \ldots, s_{\ell,\sigma}(\sqrt{\lambda_M})\right)$.

This approximation method is appealing, particularly because the eigendecomposition of the Laplacian subject to Dirichlet boundary conditions is available in closed form and is entirely independent of the covariance kernel. For stationary kernels, such as the SE kernel employed here, the approximation relies on the hyperparameters solely through the spectral density $s_{\ell,\sigma}(\cdot)$, the form of which is known analytically. Consequently, computationally expensive numerical eigendecompositions are circumvented. The analytical eigenfunctions and eigenvalues can be utilized directly, maintaining a computational cost of $\mathcal{O}(NM)$ even as hyperparameters vary.

# B. Derivation of the Loss Function

In this section, we give additional details for the optimal transport-based objective presented in Equation (7). We first start deriving ELBO by pretending there is a one-to-one correspondence between generated cells and actual cells and by assuming a probabilistic modeling setting with an explicit likelihood model denoted as $p_\theta(\mathbf{X}|\Phi)$ rather than using the push-forward mapping. The ELBO derivation serves to derive the KL regularization terms used in the main objective (Equation (7)).

## B.1. Derivation of the Evidence Lower Bound

We start by assuming the standard variational inference (Hoffman et al., 2013) setting, where we possess a dataset $\mathbf{X} = \{\boldsymbol{X}_t\}_{t\in\mathcal{T}}$, $\boldsymbol{X}_t \in \mathbb{R}^{n_t \times G}$, and a set of latent variables. In our specific context, the latent variables are hierarchical. They consist of the global GP parameters $\Phi = \{\boldsymbol{A}_f, \boldsymbol{A}_\varsigma, \boldsymbol{\ell}_f, \boldsymbol{\ell}_\varsigma, \boldsymbol{\sigma}_f, \boldsymbol{\sigma}_\varsigma\}$, and potentially cell-specific times $\boldsymbol{\tau}$, though we omit $\boldsymbol{\tau}$ here for brevity as it follows the same construction. We define a variational posterior $q_\Psi(\Phi)$ to approximate the intractable true posterior $p(\Phi|\mathbf{X})$. The marginal log-likelihood of the data can be bounded using Jensen's inequality:

$$\begin{aligned}
\log p(\mathbf{X}) &= \log \int p_\theta(\mathbf{X}|\Phi)p(\Phi)\mathrm{d}\Phi \\
&= \log \int \frac{p_\theta(\mathbf{X}|\Phi)p(\Phi)}{q_\Psi(\Phi)}q_\Psi(\Phi)\mathrm{d}\Phi \\
&\geq \mathbb{E}_{q_\Psi(\Phi)}\left[\log p_\theta(\mathbf{X}|\Phi)\right] - \mathrm{KL}(q_\Psi(\Phi)\|p(\Phi)) \\
&= \mathbb{E}_{q_\Psi(\Phi)}\left[\sum_{t\in\mathcal{T}}\log p_\theta(\boldsymbol{X}_t|\Phi)\right] - \mathrm{KL}(q_\Psi(\Phi)\|p(\Phi)).
\end{aligned} \tag{8}$$

The term $\log p_\theta(\boldsymbol{X}_t|\Phi)$ represents the reconstruction likelihood. If there were a one-to-one correspondence where the $i^{\text{th}}$ observed cell $\boldsymbol{x}_{t,i}$ corresponds to the $i^{\text{th}}$ generated cell $\hat{\boldsymbol{x}}_{t,i}$ (derived from $\Phi$ and time $t$).

To make the objective function explicit, we expand the regularization term $\mathrm{KL}(q_\Psi(\Phi)\|p(\Phi))$. Based on the mean-field assumption for $q_\Psi(\Phi)$, the variational posterior factorizes across the signal $f$ and noise $\varsigma$ components, and further across the $L$ latent dimensions. We define the variational posterior distributions for the Hilbert coefficients (weights) as multivariate Gaussians with diagonal covariances, and for the kernel hyperparameters as log-normal distributions:

$$\begin{aligned}
q_\Psi(\boldsymbol{a}_{f,l}) &= \mathcal{N}(\boldsymbol{a}_{f,l}|\boldsymbol{\mu}_{f,l}, \mathrm{diag}(\boldsymbol{\nu}_{f,l}^2)) \\
q_\Psi(\ell_{f,l}) &= \mathrm{Lognormal}(\ell_{f,l}|\mu_{\ell,f,l}, \nu_{\ell,f,l}^2) \\
q_\Psi(\sigma_{f,l}) &= \mathrm{Lognormal}(\sigma_{f,l}|\mu_{\sigma,f,l}, \nu_{\sigma,f,l}^2).
\end{aligned}$$

The variational distributions for $\varsigma$ parameters $(\boldsymbol{A}_\varsigma, \boldsymbol{\ell}_\varsigma, \boldsymbol{\sigma}_\varsigma)$ are defined analogously. The joint prior for $(\boldsymbol{A}_f, \boldsymbol{\ell}_f, \boldsymbol{\sigma}_f)$ is

$$\begin{aligned}
p(\boldsymbol{A}_f, \boldsymbol{\ell}_f, \boldsymbol{\sigma}_f) &= \prod_{l=1}^{L} p(\boldsymbol{a}_{f,l}, \ell_{f,l}, \sigma_{f,l}) \\
&= \prod_{l=1}^{L} p(\boldsymbol{a}_{f,l}|\ell_{f,l}, \sigma_{f,l})p(\ell_{f,l})p(\sigma_{f,l}).
\end{aligned}$$

The joint prior for $(\boldsymbol{A}_\varsigma, \boldsymbol{\ell}_\varsigma, \boldsymbol{\sigma}_\varsigma)$ is defined analogously. Because the priors and the variational posteriors for $(\boldsymbol{A}_f, \boldsymbol{\ell}_f, \boldsymbol{\sigma}_f)$ and $(\boldsymbol{A}_\varsigma, \boldsymbol{\ell}_\varsigma, \boldsymbol{\sigma}_\varsigma)$ are independent, the total regularization term is the sum of the KL divergences for the signal and heteroscedastic noise components:

$$\mathrm{KL}(q_\Psi(\Phi)\|p(\Phi)) = \mathcal{R}_f + \mathcal{R}_\varsigma.$$

Next, we expand the signal component $\mathcal{R}_f$ (the noise component $\mathcal{R}_\varsigma$ is identical in form). Due to the hierarchical nature of the model, the prior for the weights $p(\boldsymbol{a}_{f,l}|\ell_{f,l}, \sigma_{f,l})$ is conditional on the hyperparameters via the spectral density of the kernel $k$ as described above. Consequently, using the chain rule for the KL divergence (Murphy, 2023), the KL divergence for the weights $\boldsymbol{a}_{f,l}$ is computed as an expectation over the variational distribution of the hyperparameters:

$$\mathcal{R}_f = \sum_{l=1}^{L}\left(\underbrace{\mathbb{E}_{q_\Psi(\ell_{f,l},\sigma_{f,l})}\left[\mathrm{KL}\big(q_\Psi(\boldsymbol{a}_{f,l})\|p(\boldsymbol{a}_{f,l}|\ell_{f,l},\sigma_{f,l})\big)\right]}_{\text{weights regularization}} + \underbrace{\mathrm{KL}(q_\Psi(\ell_{f,l})\|p(\ell_{f,l}))}_{\text{lengthscale regularization}} + \underbrace{\mathrm{KL}(q_\Psi(\sigma_{f,l})\|p(\sigma_{f,l}))}_{\text{variance regularization}}\right), \tag{9}$$

where we have used the chain rule for the KL divergence for the first term. We can now write the explicit analytical forms for these terms.

**Hyperparameter KL.** Since both the priors and variational distributions for hyperparameters are log-normal, the KL divergence corresponds to that of the underlying normal distributions on the log-scale. For kernel hyperparameters $\ell$ and $\sigma$ with prior $\text{Lognormal}(0, 1)$:

$$\text{KL}(q_\Psi(\ell)\|p(\ell)) = \frac{1}{2}\left(\nu_\ell^2 + \mu_\ell^2 - 1 - \log(\nu_\ell^2)\right),$$

$$\text{KL}(q_\Psi(\sigma)\|p(\sigma)) = \frac{1}{2}\left(\nu_\sigma^2 + \mu_\sigma^2 - 1 - \log(\nu_\sigma^2)\right).$$

**Weights KL.** The KL divergence between the variational Gaussian $q_\Psi(\boldsymbol{a}) = \mathcal{N}(\boldsymbol{\mu}, \text{diag}(\boldsymbol{\nu}^2))$ and the prior $p(\boldsymbol{a}) = \mathcal{N}(\boldsymbol{0}, \boldsymbol{S}_{\ell,\sigma})$ (where $\boldsymbol{S}_{\ell,\sigma}$ is the diagonal spectral density matrix depending on sampled $\ell, \sigma$) is given by:

$$\text{KL}(q_\Psi(\boldsymbol{a})\|p(\boldsymbol{a})) = \frac{1}{2}\sum_{m=1}^{M}\left(\frac{\nu_m^2 + \mu_m^2}{[\boldsymbol{S}_{\ell,\sigma}]_{mm}} - 1 + \log[\boldsymbol{S}_{\ell,\sigma}]_{mm} - \log\nu_m^2\right).$$

In practice, we estimate the expectation w.r.t $q_\Psi(\ell, \sigma)$ in Equation (9) using Monte Carlo sampling via the reparameterization trick.

**Explicit ELBO.** Substituting these expanded terms into Equation (8), the fully explicit ELBO (before substituting the Wasserstein proxy) is:

$$\begin{aligned}
\text{ELBO} = &\mathbb{E}_{q_\Psi(A_f, A_\varsigma)}\left[\sum_{t\in\mathcal{T}}\sum_{i=1}^{n_t}\mathbb{E}_{\boldsymbol{\xi}_{t,i}\sim\mathcal{N}(\boldsymbol{0},\mathbf{I})}\left[\log p_\theta\left(\boldsymbol{x}_{t,i}\Big|\boldsymbol{z}_i(t) = \underbrace{\boldsymbol{A}_f\boldsymbol{\phi}(t)}_{\tilde{\boldsymbol{f}}(t)} + \underbrace{\exp(\boldsymbol{A}_\varsigma\boldsymbol{\phi}(t))}_{\tilde{\boldsymbol{\varsigma}}(t)}\odot\boldsymbol{\xi}_{t,i}\right)\right]\right] \\
&- \sum_{j\in\{f,\varsigma\}}\sum_{l=1}^{L}\left[\mathbb{E}_{q_\Psi(\ell_{j,l},\sigma_{j,l})}\left[\frac{1}{2}\sum_{m=1}^{M}\left(\frac{\nu_{j,l,m}^2 + \mu_{j,l,m}^2}{s_{j,l,m}} + \log s_{j,l,m} - \log\nu_{j,l,m}^2 - 1\right)\right]\right. \\
&\left. + \frac{1}{2}(\nu_{\ell,j,l}^2 + \mu_{\ell,j,l}^2 - \log\nu_{\ell,j,l}^2 - 1) + \frac{1}{2}(\nu_{\sigma,j,l}^2 + \mu_{\sigma,j,l}^2 - \log\nu_{\sigma,j,l}^2 - 1)\right],
\end{aligned}$$
(10)

where $s_{j,l,m} = [\boldsymbol{S}_{\ell_{j,l},\sigma_{j,l}}]_{mm}$ represents the spectral density of the $m^{\text{th}}$ basis function for dimension $l$ for $j\in\{f,\varsigma\}$.

## B.2. Combining Optimal Transport with KL Regularization

Equation (10) relies on the assumption that the $i^{\text{th}}$ generated sample $\hat{\boldsymbol{x}}_{t,i}$ corresponds directly to the $i^{\text{th}}$ observed sample $\boldsymbol{x}_{t,i}$. However, due to the destructive nature of scRNA-seq technology, transcriptional profiles of individual cells cannot be followed over time because each cell can be measured only once. Therefore, methods for analyzing temporal scRNA-seq data are restricted to modeling temporal population dynamics over time $t$, rather than encoding specific cell identities. Consequently, there is no inherent one-to-one mapping between the generated cells and the actual observed cells at any given time point $t$. The generated set $\hat{\boldsymbol{X}}_t$ and observed set $\boldsymbol{X}_t$ represent samples from the predicted distribution $\hat{\rho}_t$ and the data distribution $\rho_t$, respectively, but they are unordered relative to one another and may contain different numbers of samples.

Due to this lack of correspondence, we cannot evaluate the reconstruction quality using the standard element-wise log-likelihood $\sum_i \log p_\theta(\boldsymbol{x}_{t,i}|\boldsymbol{z}_{t,i})$. Instead, we require a loss function that assesses the discrepancy between the two population distributions, $\rho_t$ and $\hat{\rho}_t$.

Optimal transport offers a geometrically grounded framework to address this challenge. It generalizes the notion of reconstruction error from individual points to full probability distributions. The Wasserstein distance effectively finds the minimum cost required to transform one probability distribution into another:

$$W_2^2(\rho_t, \hat{\rho}_t) = \inf_{\pi\in\Pi(\rho_t,\hat{\rho}_t)}\int_{\mathcal{X}^2}\|\boldsymbol{x} - \hat{\boldsymbol{x}}\|_2^2\,\mathrm{d}\pi(\boldsymbol{x}, \hat{\boldsymbol{x}}).$$

Therefore, we opt to utilize an optimal transport-based objective instead of element-wise log-likelihood-based ELBO to match the population generated by the model with the observed population. To ensure computational efficiency and differentiability during training, we employ the entropic regularization discussed in Section 3. We incorporate KL regularization terms implied by the GP prior under the Hilbert space parameterization into the optimal transport framework, yielding the final objective function presented in Equation (7):

$$\mathcal{L}(\theta, \Psi) = \underbrace{\mathbb{E}_{q_\Psi(\Phi)} \left[ \sum_{t \in \mathcal{T}} W_{p,\gamma}^p (\rho_t, \hat{\rho}_t) \right]}_{\text{Expected transport cost}} + \underbrace{\text{KL}(q_\Psi(\Phi) \| p(\Phi))}_{\text{Regularization}}.$$

This formulation allows us to learn the global temporal dynamics by minimizing the transport cost between the predicted and observed cell populations at every time point, regularized by the Hilbert space GP prior.

The justification for using optimal transport loss with KL regularization is twofold. First, the transport cost is computed in expectation over the variational distribution of the GP parameters, so the transport cost remains internally consistent with the uncertainty and smoothness of the latent GP structure. Second, this objective can be understood through the lens of the WAE (Tolstikhin et al., 2018), which establishes that minimizing a transport cost in observation space combined with a latent regularization term constitutes a valid approach to generative modeling.

## C. Experimental Details

### C.1. Preprocessing

We use the preprocessing pipeline provided by Zhang et al. (2024). First, batch effects are removed across different time points to ensure temporal consistency. Dimensionality is subsequently reduced by selecting the top 2,000 highly variable genes (HVGs) detected by Scanpy (Wolf et al., 2018) and Seurat (Hao et al., 2021). To strictly avoid data leakage, these HVGs are identified using only the cells belonging to the training time points. To mitigate technical variations in sequencing depth and remove cell-specific biases, the raw gene expression counts are first normalized by the total library size. For a cell $i$, the raw count vector $x_i^{\text{raw}}$ is divided by its total counts and scaled to a fixed sum of $10^4$:

$$x_i^{\text{normalized}} = \frac{x_i^{\text{raw}}}{\sum_{g=1}^G x_{i,g}^{\text{raw}}} \times 10^4. \tag{11}$$

Subsequently, a log-transformation is applied to these normalized values to stabilize variance. A pseudo-count of 1 is included to preserve zero entries, resulting in the final processed expression vector:

$$x_i = \log(x_i^{\text{normalized}} + 1). \tag{12}$$

*Table 2.* Number of observed cells at each time point for the ZB, DR, and SC datasets. Time points are indexed starting from $t = 0$. Dashes ($-$) indicate that the dataset does not contain measurements for that specific time point index.

| Dataset | 0 | 1 | 2 | 3 | 4 | 5 | 6 | 7 | 8 | 9 | 10 | 11 | 12 | 13 | 14 | 15 | 16 | 17 | 18 |
|---------|---|---|---|---|---|---|---|---|---|---|----|----|----|----|----|----|----|----|----|
| **ZB** | 311 | 200 | 1,158 | 1,467 | 5,716 | 1,026 | 4,101 | 6,178 | 5,442 | 7,114 | 1,614 | 4,404 | – | – | – | – | – | – | – |
| **DR** | 3,944 | 2,801 | 947 | 3,139 | 2,074 | 2,147 | 3,311 | 1,692 | 1,856 | 1,491 | 3,984 | – | – | – | – | – | – | – | – |
| **SC** | 800 | 560 | 1,371 | 1,413 | 1,608 | 1,377 | 1,153 | 1,156 | 2,420 | 886 | 732 | 712 | 747 | 708 | 1,444 | 2,061 | 2,106 | 1,622 | 743 |

*Table 3.* Dataset details and leave-out time points for the three evaluation tasks.

| Dataset | # TPs | Leave-Out Time Points | | | Raw Data | Preprocessed Data |
|---------|-------|-----------|-------------|-------------|----------|-------------------|
| | | **Easy Task** | **Medium Task** | **Hard Task** | | |
| **ZB** | 12 | 4, 6, 8 | 10, 11 | 2, 4, 6, 8, 10, 11 | SCP162 | |
| **DR** | 11 | 4, 6, 8 | 8, 9, 10 | 2, 4, 6, 8, 9, 10 | GSE190149 | Figshare (Zhang et al., 2024) |
| **SC** | 19 | 5, 10, 15 | 16, 17, 18 | 5, 7, 9, 11, 15, 16, 17, 18 | WOT Tutorial | |

## C.2. Hyperparameters, Training and Evaluation

**Implementation and Evaluation.**   The framework was implemented using PyTorch (Paszke et al., 2019). During the training phase, the objective function was computed using the Sinkhorn algorithm provided by the GeomLoss package (Feydy et al., 2019), with the blur and scaling parameters set to $0.05$ ($\gamma = 0.05^2$ in Equation (1)) and $0.5$, respectively. Conversely, for quantitative evaluation, 2-Wasserstein distance ($W_2$) was calculated using the POT ("emd2" function) package (Flamary et al., 2021) on $\mathcal{X}$ to ensure a precise metric devoid of entropic regularization bias.

**LGP-OT.**   The hyperparameter search space for LGP-OT was defined as follows: the latent dimensionality $L \in \{32, 50\}$; $h_\theta$ network architecture $\in \{\text{None}, [50], [50, 50]\}$ with ReLU activation function; the number of Hilbert space basis functions $M \in \{5, 6, 7, 8, 9, 10\}$; and the boundary factor $\frac{J}{\max(\mathcal{T}^{\text{standardized}})} \in \{1.5, 2.0, 2.5, 3.0\}$. We used the same number of basis functions and the boundary factor for both signal and noise models. The optimal configuration was identified via 3-fold cross-validation on the training data, adhering to the same protocol used for the baselines. We fix the standard deviation $\delta$ of the model variant with latent cell-time modeling (Section 4.3) to $0.1$. We specify the categorical kernel used in the model variant defined in Section 4.4 as a diagonal kernel with unit entries.

Following hyperparameter selection, the final model was trained for a maximum of $10,000$ iterations using the Adam optimizer (Kingma, 2014) with an initial learning rate of $0.001$ and a batch size of $256$ samples per training time point. During this final training phase, a validation subset comprising $5\%$ of the data from three randomly sampled training time points was reserved solely for monitoring convergence. An adaptive scheduler decayed the learning rate by a factor of $0.5$ if this validation loss did not improve for $200$ iterations, and early stopping was applied to prevent overfitting. The final model weights were selected from the checkpoint achieving the lowest loss on this $5\%$ validation subset.

**Baselines.**   To ensure a rigorous and unbiased evaluation, hyperparameter search for all baseline models was performed using the same 3-fold cross-validation protocol applied to LGP-OT. While the specific search spaces and architectural choices were adopted from established benchmarks (Zhang et al., 2024) and original publications, all models were trained until convergence using their recommended optimization protocols (optimizers, learning rates, and schedulers). Consistent with their original implementations, PRESCIENT (Yeo et al., 2021), PI-SDE (Jiang & Wan, 2024), and VGFM (Wang et al., 2025) train a dynamics model in a 50-dimensional PCA space. Whereas, MIOFlow (Huguet et al., 2022) combines PCA and the geodesic autoencoder to model the high-dimensional data in a pre-trained latent space. For these models, all predicted distributions were projected back to the original observation space ($\mathcal{X}$) before evaluation.

**scNODE.** The latent dimensionality was fixed at $50$, as suggested in (Zhang et al., 2024). The network architecture was optimized by varying the configuration of the encoder, decoder, and drift networks across the set $\{\text{None}, [50], [50, 50]\}$. The dynamic regularization coefficient was tuned within the range $[0.0, 10.0]$.

**PRESCIENT.** Hyperparameter tuning focused on the standard deviation of Gaussian noise ($[0.0, 1.0]$), the regularization strength ($[0.0, 0.1]$), and the gradient clipping threshold ($[0.0, 1.0]$). Additionally, the depth of the hidden layers was selected from the set $\{1, 2, 3\}$.

**MIOFlow.** The hyperparameter search space included geodesic autoencoder embedding dimensions of $\{10, 50, 100\}$. The partition encoder architecture was selected from $\{[50, 100], [50, 100, 100]\}$, while the ODE solver network layers were varied across $\{[16], [16, 16], [16, 32, 16]\}$. The regularization coefficient for the density loss was tuned within the range $[1.0, 100.0]$.

**PI-SDE.** We followed the hyperparameter configuration detailed in the original publication (Jiang & Wan, 2024). Specifically, the potential function was parameterized by a fully connected 2-layer network with $400$ units and softplus activation. The diffusion coefficient was selected among three options: modeled by a separate neural network with the same architecture as the potential function, a constant value of $0.1$, or a vector parameter to be optimized.

**VGFM.** We followed the hyperparameter selection strategy presented in the original publication (Wang et al., 2025). Both the velocity field and growth function are parameterized by a fully connected 5-layer network with $256$ units and LeakyReLU activation. To select the entropy regularization parameter, the relaxation coefficient is first set to a constant value of $50$, then a grid search is applied by increasing the entropy regularization parameter from $0.01$ up until numerical stability is achieved. After that, the relaxation coefficient is determined similarly to the elbow method used in clustering by gradually increasing in the range of $[0.001, 10]$. The optimal value is selected according to the stabilization point of the curve. While VGFM demonstrated to work with high-dimensional inputs (Wang et al., 2025), we utilized 50-dimensional PCA as it yielded better

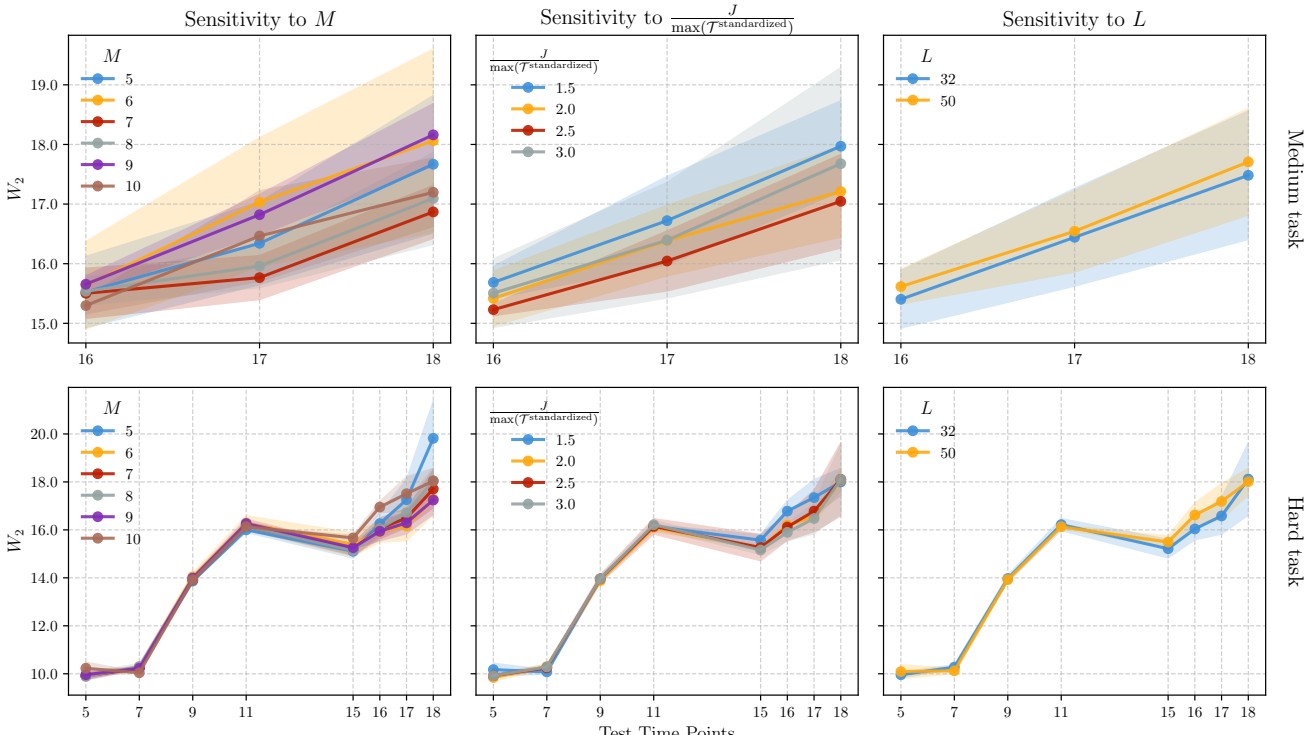

*Figure 5.* Sensitivity analysis to key hyperparameters on the SC dataset. Results are shown for medium (top) and hard (bottom) tasks using the validation splits from hyperparameter tuning.

results for this baseline.

**Homoscedastic ablations.** Finally, to evaluate the impact of time-varying noise, we tested two homoscedastic versions of our model: we either fixed the noise standard deviation to $0.1$ for all latent dimensions ($\varsigma_l(t) = 0.1$) or treated it as a vector of optimizable parameters ($\varsigma(t) = \varsigma \in \mathbb{R}^L_+$).

All training was performed on a single NVIDIA Tesla V100 GPU with 32GB of memory.

### C.3. Sensitivity of LGP-OT

To assess the robustness of our approach to key hyperparameters of the latent GP model, we conducted sensitivity analyses with respect to the number of basis functions $M$, the boundary factor $\frac{J}{\max(\mathcal{T}^{\text{standardized}})}$, and the latent dimensionality. These experiments were performed using the same cross-validation splits employed during hyperparameter tuning, where each model is trained on approximately $2/3$ of the available training data. For each hyperparameter value, the reported results aggregate performance across all corresponding configurations during hyperparameter tuning. As shown in Figure 5, performance exhibits smooth and consistent behavior across a broad range of hyperparameter values in both medium and hard extrapolation settings, suggesting that the proposed framework is not overly sensitive to these design choices. Importantly, this analysis is intended to assess qualitative stability trends rather than absolute performance. All results reported in the main experiments are obtained by retraining the model on the full training set using the selected hyperparameters.

### C.4. Perturbation Analysis

We update the original variational parameters $\Psi$ with respect to perturbed genes using different levels of perturbation, as denoted in Section 5.1. We fix the $\eta$ parameter to $0.001$ for all perturbations. We perform the optimization for $100$ iterations.

The neural network architecture and hyperparameters of the cell type classifier are presented in Table 4, which is trained using the Adam optimizer with a learning rate of $0.001$ for $500$ epochs on the terminal point observed data with class labels. To validate the external classifier used in the perturbation evaluation, we report its performance under 5-fold stratified

cross-validation using pooled out-of-fold predictions. Overall performance of the classifier is provided in Table 5. Table 6 and the row-normalized confusion matrix in Figure 7 further show the strong performance of the classifier for the primary lineages targeted (PSM and Hindbrain) in the perturbation experiment. To classify the generated cells by LGP-OT, the classifier is trained on all available cells at the terminal time point.

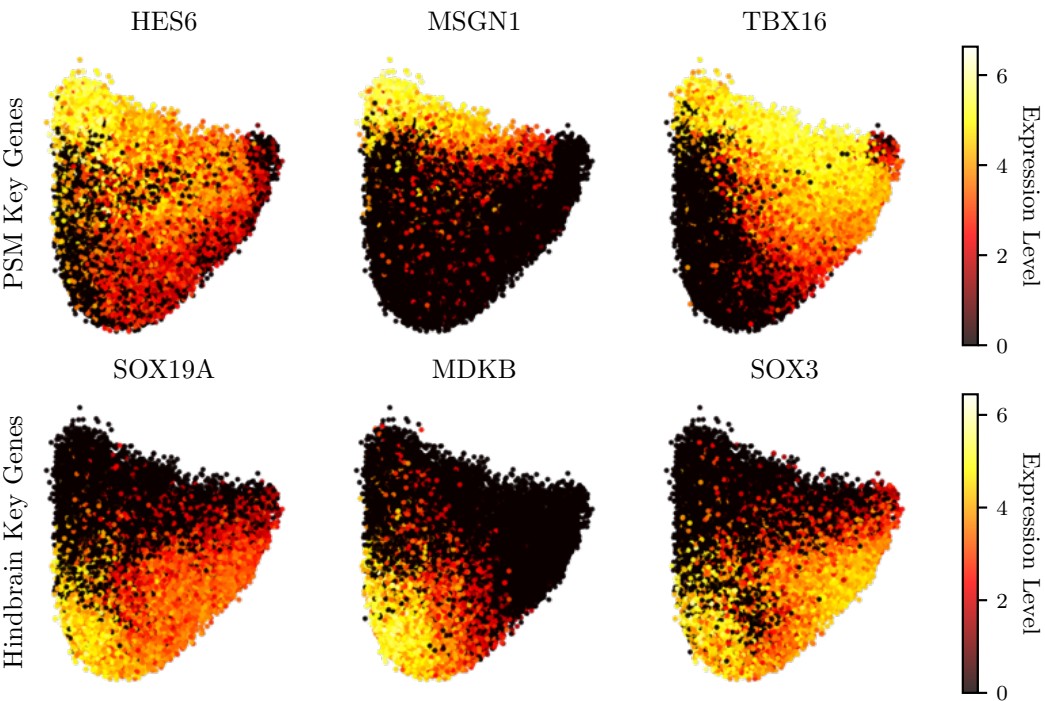

*Figure 6.* PCA visualization of the most DE genes of PSM and Hindbrain cells identified from the final time point of the ZB dataset.

*Table 4.* Classifier neural network architecture used in perturbation analysis.

| Hyperparameter | Value |
| --- | --- |
| Hidden layer sizes | 64, 32 |
| Hidden layer activation | ReLU |
| Output layer activation | Softmax |
| L2 regularization | 0.0001 |

*Table 5.* Overall performance of the classifier from pooled out-of-fold predictions under 5-fold stratified cross-validation.

| Accuracy | Balanced Accuracy | Macro-Precision | Macro-Recall | Macro-F1 |
| --- | --- | --- | --- | --- |
| $0.8610 \pm 0.0039$ | $0.7875 \pm 0.0126$ | $0.8441 \pm 0.0347$ | $0.7875 \pm 0.0126$ | $0.8020 \pm 0.0155$ |

## C.5. Full Results

*Table 6.* Precision, recall and F1 for each cell type from pooled out-of-fold predictions under 5-fold stratified cross-validation. Support denotes the number of test cells aggregated across folds.

| Cell type | Support | Precision | Recall | F1 |
|---|---|---|---|---|
| PSM | 540 | 0.9019 | 0.9537 | 0.9271 |
| Hindbrain | 511 | 0.7871 | 0.8611 | 0.8224 |
| Optic Cup | 302 | 0.9003 | 0.9570 | 0.9278 |
| Neural | 300 | 0.9070 | 0.9100 | 0.9085 |
| Spinal | 265 | 0.8132 | 0.8377 | 0.8253 |
| NAN | 262 | 0.5181 | 0.3817 | 0.4396 |
| Placode | 249 | 0.9044 | 0.9116 | 0.9080 |
| Somites | 241 | 0.9360 | 0.9710 | 0.9532 |
| Midbrain | 220 | 0.8609 | 0.9000 | 0.8800 |
| Epidermis | 199 | 0.8190 | 0.9095 | 0.8619 |
| Tailbud | 166 | 0.8075 | 0.7831 | 0.7951 |
| Dorsal Diencephalon | 157 | 0.8790 | 0.8790 | 0.8790 |
| Heart Primordium | 156 | 0.9423 | 0.9423 | 0.9423 |
| Endoderm | 148 | 0.9583 | 0.9324 | 0.9452 |
| Telencephalon | 96 | 0.8646 | 0.8646 | 0.8646 |
| Cephalic Mesoderm | 93 | 0.9789 | 1.0000 | 0.9894 |
| Diencephalon Ventral | 92 | 0.9412 | 0.8696 | 0.9040 |
| Adaxial Cells | 91 | 0.9333 | 0.7692 | 0.8434 |
| Notochord | 78 | 0.9459 | 0.8974 | 0.9211 |
| Hematopoeitic | 61 | 0.8033 | 0.8033 | 0.8033 |
| Prechordal Plate | 45 | 1.0000 | 0.9778 | 0.9888 |
| Floor Plate | 40 | 0.7241 | 0.5250 | 0.6087 |
| Pronephros | 33 | 0.8621 | 0.7576 | 0.8065 |
| Integument | 25 | 0.8824 | 0.6000 | 0.7143 |
| EVL/Periderm | 16 | 1.0000 | 0.4375 | 0.6087 |
| YSL | 14 | 0.5000 | 0.0714 | 0.1250 |
| Primordial Germ Cells | 4 | 1.0000 | 0.5000 | 0.6667 |

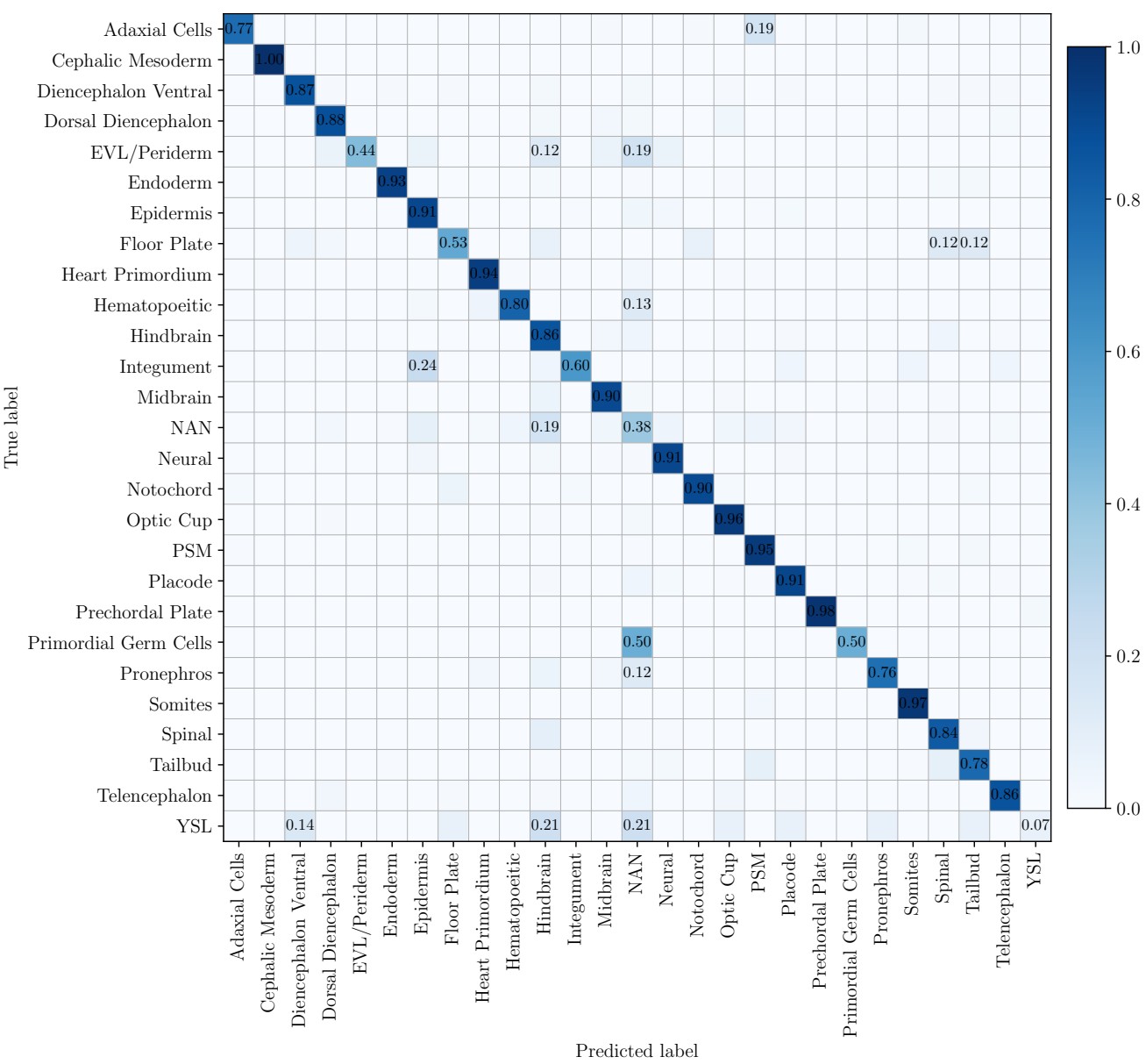

*Figure 7.* Normalized confusion matrix of the classifier from pooled out-of-fold predictions under 5-fold stratified cross-validation.

*Table 7.* Performance comparison on easy (interpolation) and medium (extrapolation) tasks across three datasets.

| Model | ZB/Easy | | | ZB/Medium | |
|---|---|---|---|---|---|
| | $t=4$ | $t=6$ | $t=8$ | $t=10$ | $t=11$ |
| LGP-OT | $29.69 \pm_{0.11}$ | $\mathbf{27.07} \pm_{\mathbf{0.05}}$ | $\mathbf{29.59} \pm_{\mathbf{0.07}}$ | $\mathbf{33.74} \pm_{\mathbf{0.02}}$ | $36.35 \pm_{0.44}$ |
| LGP-OT (w. time) | $\mathbf{29.39} \pm_{\mathbf{0.07}}$ | $27.33 \pm_{0.08}$ | $29.96 \pm_{0.18}$ | $33.88 \pm_{0.06}$ | $\mathbf{35.36} \pm_{\mathbf{0.04}}$ |
| LGP-OT ($\varsigma(t)=0.1$) | $31.07 \pm_{0.17}$ | $28.69 \pm_{0.26}$ | $31.17 \pm_{0.23}$ | $34.64 \pm_{0.20}$ | $36.27 \pm_{0.40}$ |
| LGP-OT ($\varsigma(t)=\varsigma$) | $31.44 \pm_{0.20}$ | $29.38 \pm_{0.16}$ | $32.06 \pm_{0.08}$ | $35.70 \pm_{0.31}$ | $38.47 \pm_{0.53}$ |
| VGFM | $30.02 \pm_{0.28}$ | $27.57 \pm_{0.18}$ | $30.27 \pm_{0.27}$ | $34.39 \pm_{0.40}$ | $36.42 \pm_{0.62}$ |
| scNODE | $29.58 \pm_{0.12}$ | $28.23 \pm_{0.14}$ | $30.93 \pm_{0.11}$ | $35.31 \pm_{0.15}$ | $37.66 \pm_{0.18}$ |
| PI-SDE | $29.67 \pm_{0.21}$ | $28.75 \pm_{0.12}$ | $31.50 \pm_{0.14}$ | $36.89 \pm_{0.52}$ | $39.48 \pm_{0.76}$ |
| MIOFlow | $29.71 \pm_{0.10}$ | $29.05 \pm_{0.17}$ | $31.76 \pm_{0.18}$ | $36.19 \pm_{0.64}$ | $38.22 \pm_{0.92}$ |
| PRESCIENT | $40.97 \pm_{0.40}$ | $36.73 \pm_{0.55}$ | $36.99 \pm_{0.75}$ | $47.13 \pm_{2.17}$ | $48.28 \pm_{2.12}$ |

| Model | DR/Easy | | | DR/Medium | | |
|---|---|---|---|---|---|---|
| | $t=4$ | $t=6$ | $t=8$ | $t=8$ | $t=9$ | $t=10$ |
| LGP-OT | $26.35 \pm_{0.06}$ | $28.22 \pm_{0.02}$ | $30.74 \pm_{0.03}$ | $32.15 \pm_{0.04}$ | $32.52 \pm_{0.08}$ | $36.51 \pm_{0.04}$ |
| LGP-OT (w. type) | $26.46 \pm_{0.08}$ | $28.18 \pm_{0.04}$ | $\mathbf{30.59} \pm_{\mathbf{0.05}}$ | $31.92 \pm_{0.06}$ | $31.79 \pm_{0.10}$ | $36.48 \pm_{0.06}$ |
| LGP-OT (w. time) | $26.25 \pm_{0.13}$ | $28.32 \pm_{0.09}$ | $30.82 \pm_{0.05}$ | $32.24 \pm_{0.11}$ | $\mathbf{31.54} \pm_{\mathbf{0.04}}$ | $36.33 \pm_{0.05}$ |
| LGP-OT (w. time + type) | $\mathbf{26.09} \pm_{\mathbf{0.03}}$ | $\mathbf{28.12} \pm_{\mathbf{0.04}}$ | $\mathbf{30.59} \pm_{\mathbf{0.06}}$ | $\mathbf{31.88} \pm_{\mathbf{0.07}}$ | $31.63 \pm_{0.09}$ | $\mathbf{36.32} \pm_{\mathbf{0.20}}$ |
| LGP-OT ($\varsigma(t)=0.1$) | $26.85 \pm_{0.19}$ | $29.16 \pm_{0.25}$ | $31.90 \pm_{0.28}$ | $33.24 \pm_{0.17}$ | $33.05 \pm_{0.20}$ | $37.41 \pm_{0.22}$ |
| LGP-OT ($\varsigma(t)=\varsigma$) | $27.96 \pm_{0.29}$ | $29.85 \pm_{0.10}$ | $32.92 \pm_{0.10}$ | $34.73 \pm_{0.29}$ | $35.49 \pm_{0.37}$ | $37.90 \pm_{0.22}$ |
| VGFM | $\mathbf{26.09} \pm_{\mathbf{0.19}}$ | $28.34 \pm_{0.07}$ | $31.29 \pm_{0.13}$ | $33.73 \pm_{0.04}$ | $32.88 \pm_{0.26}$ | $37.06 \pm_{0.47}$ |
| scNODE | $26.58 \pm_{0.12}$ | $28.65 \pm_{0.11}$ | $31.46 \pm_{0.15}$ | $33.24 \pm_{0.16}$ | $33.67 \pm_{0.38}$ | $38.04 \pm_{0.35}$ |
| MIOFlow | $26.44 \pm_{0.12}$ | $28.55 \pm_{0.06}$ | $31.65 \pm_{0.13}$ | $33.60 \pm_{0.12}$ | $34.49 \pm_{0.24}$ | $38.76 \pm_{0.82}$ |
| PI-SDE | $26.70 \pm_{0.03}$ | $28.76 \pm_{0.04}$ | $31.39 \pm_{0.07}$ | $32.80 \pm_{0.09}$ | $33.49 \pm_{0.37}$ | $38.27 \pm_{0.57}$ |
| PRESCIENT | $30.16 \pm_{0.09}$ | $31.16 \pm_{0.08}$ | $32.48 \pm_{0.07}$ | $34.66 \pm_{0.24}$ | $33.26 \pm_{0.22}$ | $37.64 \pm_{0.34}$ |

| Model | SC/Easy | | | SC/Medium | | |
|---|---|---|---|---|---|---|
| | $t=5$ | $t=10$ | $t=15$ | $t=16$ | $t=17$ | $t=18$ |
| LGP-OT | $\mathbf{10.19} \pm_{\mathbf{0.05}}$ | $15.95 \pm_{0.04}$ | $\mathbf{14.32} \pm_{\mathbf{0.04}}$ | $\mathbf{15.24} \pm_{\mathbf{0.04}}$ | $15.95 \pm_{0.31}$ | $\mathbf{16.17} \pm_{\mathbf{0.36}}$ |
| LGP-OT (w. time) | $10.24 \pm_{0.08}$ | $\mathbf{15.88} \pm_{\mathbf{0.06}}$ | $14.49 \pm_{0.10}$ | $15.70 \pm_{0.26}$ | $16.03 \pm_{0.32}$ | $16.20 \pm_{0.28}$ |
| LGP-OT ($\varsigma(t)=0.1$) | $10.90 \pm_{0.34}$ | $16.81 \pm_{0.36}$ | $15.63 \pm_{0.41}$ | $16.69 \pm_{0.37}$ | $16.86 \pm_{0.49}$ | $17.17 \pm_{0.54}$ |
| LGP-OT ($\varsigma(t)=\varsigma$) | $11.02 \pm_{0.08}$ | $18.22 \pm_{0.29}$ | $18.46 \pm_{0.19}$ | $18.17 \pm_{0.07}$ | $18.70 \pm_{0.16}$ | $18.21 \pm_{0.30}$ |
| VGFM | $10.35 \pm_{0.20}$ | $16.72 \pm_{0.51}$ | $16.59 \pm_{1.21}$ | $18.14 \pm_{1.00}$ | $17.79 \pm_{0.78}$ | $18.54 \pm_{0.77}$ |
| scNODE | $11.09 \pm_{0.23}$ | $16.99 \pm_{0.39}$ | $16.15 \pm_{0.52}$ | $16.30 \pm_{0.30}$ | $\mathbf{15.94} \pm_{\mathbf{0.20}}$ | $16.44 \pm_{0.17}$ |
| MIOFlow | $10.70 \pm_{0.12}$ | $17.03 \pm_{0.38}$ | $17.36 \pm_{0.61}$ | $17.58 \pm_{0.27}$ | $17.06 \pm_{0.24}$ | $17.26 \pm_{0.38}$ |
| PI-SDE | $10.68 \pm_{0.11}$ | $17.07 \pm_{0.33}$ | $14.91 \pm_{0.34}$ | $18.44 \pm_{0.87}$ | $17.88 \pm_{0.98}$ | $18.89 \pm_{1.03}$ |
| PRESCIENT | $14.48 \pm_{0.10}$ | $16.77 \pm_{0.09}$ | $17.88 \pm_{0.20}$ | $18.59 \pm_{0.65}$ | $17.80 \pm_{0.62}$ | $17.98 \pm_{0.64}$ |

*Table 8.* Performance comparison on hard tasks (interpolation + extrapolation) across three datasets.

| Model | ZB/Hard | | | | | |
|---|---|---|---|---|---|---|
| | $t = 2$ | $t = 4$ | $t = 6$ | $t = 8$ | $t = 10$ | $t = 11$ |
| LGP-OT | **34.01** $\pm_{0.04}$ | 31.98 $\pm_{0.12}$ | **28.70** $\pm_{0.03}$ | **30.85** $\pm_{0.01}$ | **34.90** $\pm_{0.02}$ | 36.79 $\pm_{0.03}$ |
| LGP-OT (w. time) | 34.13 $\pm_{0.11}$ | **31.52** $\pm_{0.08}$ | 28.96 $\pm_{0.10}$ | 31.20 $\pm_{0.16}$ | 35.01 $\pm_{0.11}$ | **36.47** $\pm_{0.15}$ |
| LGP-OT ($\varsigma(t) = 0.1$) | 35.87 $\pm_{0.28}$ | 33.33 $\pm_{0.29}$ | 30.36 $\pm_{0.37}$ | 32.52 $\pm_{0.38}$ | 35.73 $\pm_{0.15}$ | 37.37 $\pm_{0.10}$ |
| LGP-OT ($\varsigma(t) = \varsigma$) | 34.79 $\pm_{0.05}$ | 33.83 $\pm_{0.21}$ | 31.25 $\pm_{0.08}$ | 33.50 $\pm_{0.06}$ | 36.66 $\pm_{0.12}$ | 38.60 $\pm_{0.47}$ |
| VGFM | 34.03 $\pm_{0.08}$ | 32.17 $\pm_{0.06}$ | 29.11 $\pm_{0.15}$ | 31.10 $\pm_{0.20}$ | 35.64 $\pm_{0.25}$ | 40.19 $\pm_{0.54}$ |
| scNODE | 34.08 $\pm_{0.16}$ | 32.04 $\pm_{0.18}$ | 29.98 $\pm_{0.13}$ | 32.50 $\pm_{0.24}$ | 36.57 $\pm_{0.26}$ | 38.00 $\pm_{0.25}$ |
| PI-SDE | 34.04 $\pm_{0.02}$ | 31.76 $\pm_{0.20}$ | 30.51 $\pm_{0.14}$ | 32.99 $\pm_{0.37}$ | 37.34 $\pm_{0.54}$ | 39.31 $\pm_{0.80}$ |
| MIOFlow | 34.34 $\pm_{0.31}$ | 31.73 $\pm_{0.13}$ | 30.47 $\pm_{0.12}$ | 33.17 $\pm_{0.43}$ | 36.53 $\pm_{0.21}$ | 37.99 $\pm_{0.15}$ |
| PRESCIENT | 52.71 $\pm_{0.34}$ | 45.53 $\pm_{0.74}$ | 39.98 $\pm_{1.65}$ | 40.46 $\pm_{2.68}$ | 46.44 $\pm_{3.16}$ | 47.69 $\pm_{3.16}$ |

| Model | DR/Hard | | | | | |
|---|---|---|---|---|---|---|
| | $t = 2$ | $t = 4$ | $t = 6$ | $t = 8$ | $t = 9$ | $t = 10$ |
| LGP-OT | 29.40 $\pm_{0.13}$ | 30.61 $\pm_{0.07}$ | 32.33 $\pm_{0.06}$ | 33.78 $\pm_{0.02}$ | 33.85 $\pm_{0.11}$ | 38.17 $\pm_{0.20}$ |
| LGP-OT (w. type) | **28.75** $\pm_{0.04}$ | **30.39** $\pm_{0.06}$ | **32.19** $\pm_{0.01}$ | **33.75** $\pm_{0.07}$ | **33.52** $\pm_{0.04}$ | 37.82 $\pm_{0.23}$ |
| LGP-OT (w. time) | 29.59 $\pm_{0.06}$ | 30.48 $\pm_{0.06}$ | 32.30 $\pm_{0.02}$ | 33.83 $\pm_{0.11}$ | 33.68 $\pm_{0.08}$ | **37.71** $\pm_{0.08}$ |
| LGP-OT (w. time + type) | 28.84 $\pm_{0.05}$ | 30.48 $\pm_{0.07}$ | 32.20 $\pm_{0.02}$ | 33.77 $\pm_{0.08}$ | **33.52** $\pm_{0.07}$ | 37.88 $\pm_{0.30}$ |
| LGP-OT ($\varsigma(t) = 0.1$) | 30.14 $\pm_{0.21}$ | 31.28 $\pm_{0.21}$ | 33.29 $\pm_{0.10}$ | 34.93 $\pm_{0.09}$ | 34.88 $\pm_{0.14}$ | 38.79 $\pm_{0.33}$ |
| LGP-OT ($\varsigma(t) = \varsigma$) | 30.77 $\pm_{0.15}$ | 32.15 $\pm_{0.08}$ | 34.14 $\pm_{0.03}$ | 35.66 $\pm_{0.11}$ | 36.77 $\pm_{0.21}$ | 39.21 $\pm_{0.18}$ |
| VGFM | 30.27 $\pm_{0.05}$ | 31.30 $\pm_{0.06}$ | 33.36 $\pm_{0.03}$ | 35.62 $\pm_{0.06}$ | 35.36 $\pm_{0.20}$ | 38.47 $\pm_{0.23}$ |
| scNODE | 29.95 $\pm_{0.29}$ | 30.71 $\pm_{0.16}$ | 32.88 $\pm_{0.12}$ | 34.82 $\pm_{0.23}$ | 34.63 $\pm_{0.41}$ | 38.26 $\pm_{0.51}$ |
| MIOFlow | 29.74 $\pm_{0.18}$ | 30.97 $\pm_{0.07}$ | 32.84 $\pm_{0.08}$ | 35.25 $\pm_{0.17}$ | 36.78 $\pm_{0.34}$ | 41.43 $\pm_{1.07}$ |
| PI-SDE | 30.34 $\pm_{0.14}$ | 30.82 $\pm_{0.09}$ | 32.71 $\pm_{0.05}$ | 34.18 $\pm_{0.17}$ | 34.55 $\pm_{0.27}$ | 38.90 $\pm_{0.54}$ |
| PRESCIENT | 32.68 $\pm_{0.13}$ | 32.36 $\pm_{0.11}$ | 33.08 $\pm_{0.11}$ | 35.51 $\pm_{0.21}$ | 34.19 $\pm_{0.14}$ | 38.12 $\pm_{0.15}$ |

| Model | SC/Hard | | | | | | | |
|---|---|---|---|---|---|---|---|---|
| | $t = 5$ | $t = 7$ | $t = 9$ | $t = 11$ | $t = 15$ | $t = 16$ | $t = 17$ | $t = 18$ |
| LGP-OT | **9.92** $\pm_{0.02}$ | **10.27** $\pm_{0.01}$ | **13.84** $\pm_{0.01}$ | **16.12** $\pm_{0.01}$ | **14.97** $\pm_{0.01}$ | **15.46** $\pm_{0.02}$ | 15.44 $\pm_{0.05}$ | 16.29 $\pm_{0.04}$ |
| LGP-OT (w. time) | **9.92** $\pm_{0.08}$ | 10.33 $\pm_{0.08}$ | 14.11 $\pm_{0.06}$ | 16.20 $\pm_{0.08}$ | 15.48 $\pm_{0.06}$ | 15.74 $\pm_{0.10}$ | **15.05** $\pm_{0.13}$ | **15.53** $\pm_{0.08}$ |
| LGP-OT ($\varsigma(t) = 0.1$) | 11.31 $\pm_{0.38}$ | 11.60 $\pm_{0.34}$ | 15.10 $\pm_{0.36}$ | 17.69 $\pm_{0.31}$ | 16.49 $\pm_{0.40}$ | 17.44 $\pm_{0.34}$ | 19.03 $\pm_{0.31}$ | 19.68 $\pm_{0.20}$ |
| LGP-OT ($\varsigma(t) = \varsigma$) | 11.07 $\pm_{0.37}$ | 11.38 $\pm_{0.37}$ | 15.71 $\pm_{0.29}$ | 18.77 $\pm_{0.33}$ | 18.25 $\pm_{0.22}$ | 18.28 $\pm_{0.21}$ | 18.66 $\pm_{0.13}$ | 18.60 $\pm_{0.23}$ |
| VGFM | 10.80 $\pm_{0.08}$ | 11.00 $\pm_{0.13}$ | 14.75 $\pm_{0.13}$ | 18.15 $\pm_{0.36}$ | 17.06 $\pm_{0.51}$ | 17.81 $\pm_{0.78}$ | 17.54 $\pm_{0.70}$ | 18.67 $\pm_{0.88}$ |
| scNODE | 10.44 $\pm_{0.05}$ | 10.90 $\pm_{0.05}$ | 14.45 $\pm_{0.04}$ | 17.54 $\pm_{0.22}$ | 16.76 $\pm_{0.21}$ | 17.90 $\pm_{0.24}$ | 17.26 $\pm_{0.28}$ | 18.14 $\pm_{0.34}$ |
| MIOFlow | 10.52 $\pm_{0.21}$ | 11.19 $\pm_{0.25}$ | 15.10 $\pm_{0.21}$ | 18.52 $\pm_{0.82}$ | 18.01 $\pm_{0.43}$ | 18.99 $\pm_{0.74}$ | 18.93 $\pm_{1.08}$ | 19.82 $\pm_{1.48}$ |
| PI-SDE | 10.40 $\pm_{0.10}$ | 10.73 $\pm_{0.15}$ | 14.12 $\pm_{0.10}$ | 17.01 $\pm_{0.25}$ | 18.34 $\pm_{0.66}$ | 20.20 $\pm_{0.67}$ | 19.92 $\pm_{0.73}$ | 21.32 $\pm_{0.72}$ |
| PRESCIENT | 13.66 $\pm_{0.18}$ | 13.81 $\pm_{0.21}$ | 15.55 $\pm_{0.18}$ | 17.05 $\pm_{0.20}$ | 18.24 $\pm_{0.33}$ | 18.86 $\pm_{0.36}$ | 18.06 $\pm_{0.35}$ | 18.44 $\pm_{0.37}$ |

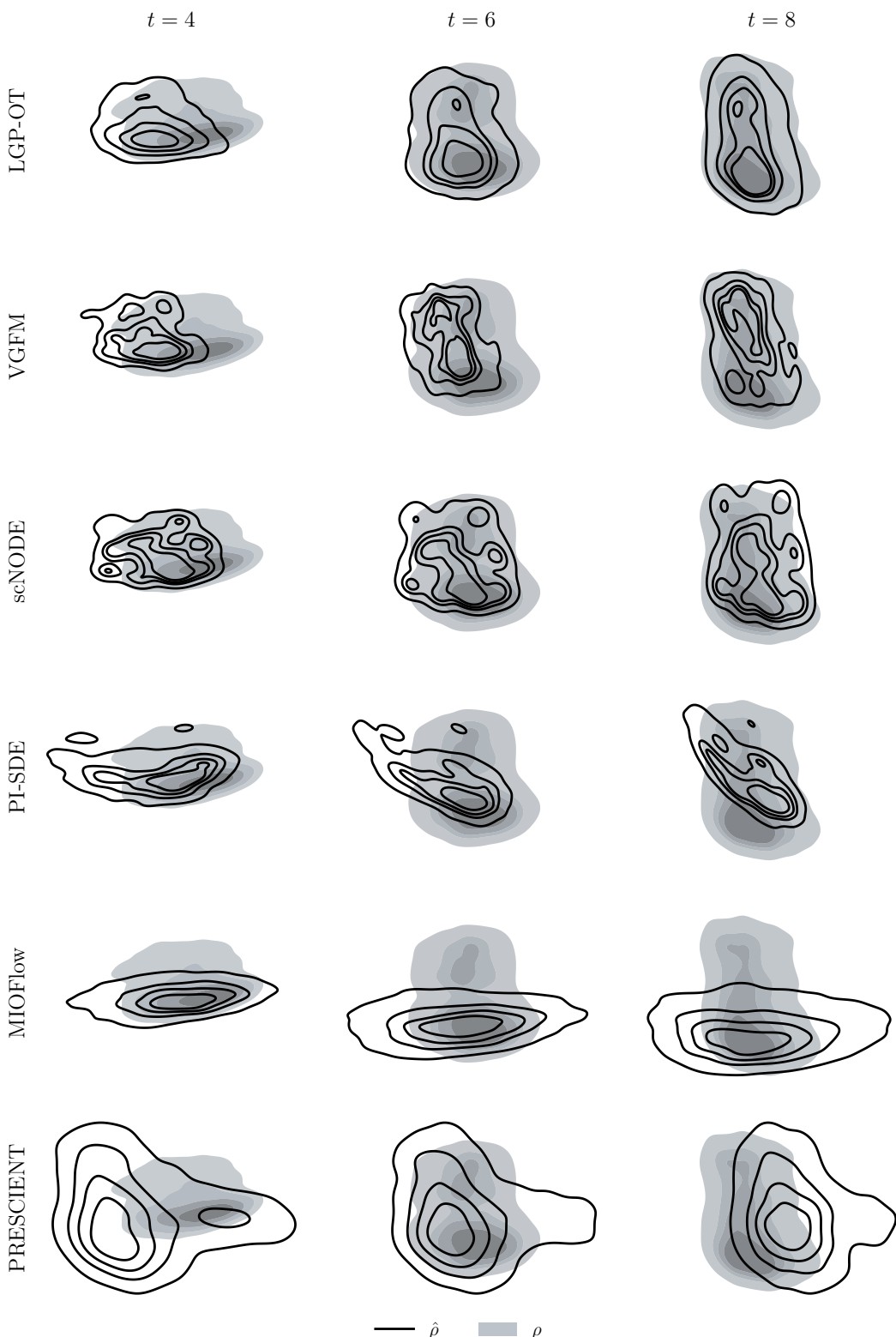

*Figure 8.* Comparison of predicted and observed distributions across models on test time points of the ZB easy task. The black contours, $\hat{\rho}$, represent the predicted cell density distributions generated by each dynamical model, while the grey shaded regions, $\rho$, depict the observed cell density.

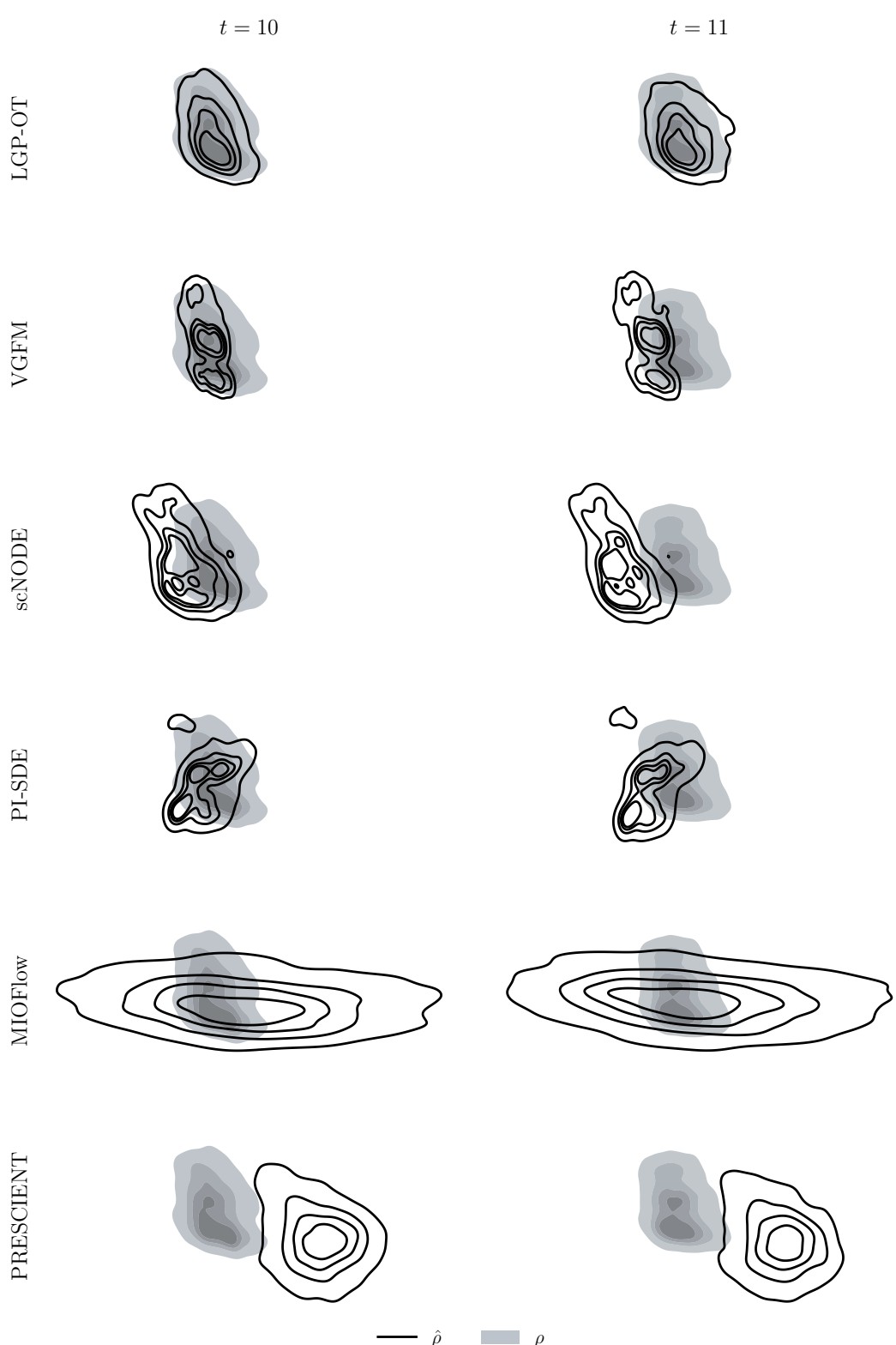

*Figure 9.* Comparison of predicted and observed distributions across models on test time points of the ZB medium task. The black contours, $\hat{\rho}$, represent the predicted cell density distributions generated by each dynamical model, while the grey shaded regions, $\rho$, depict the observed cell density.

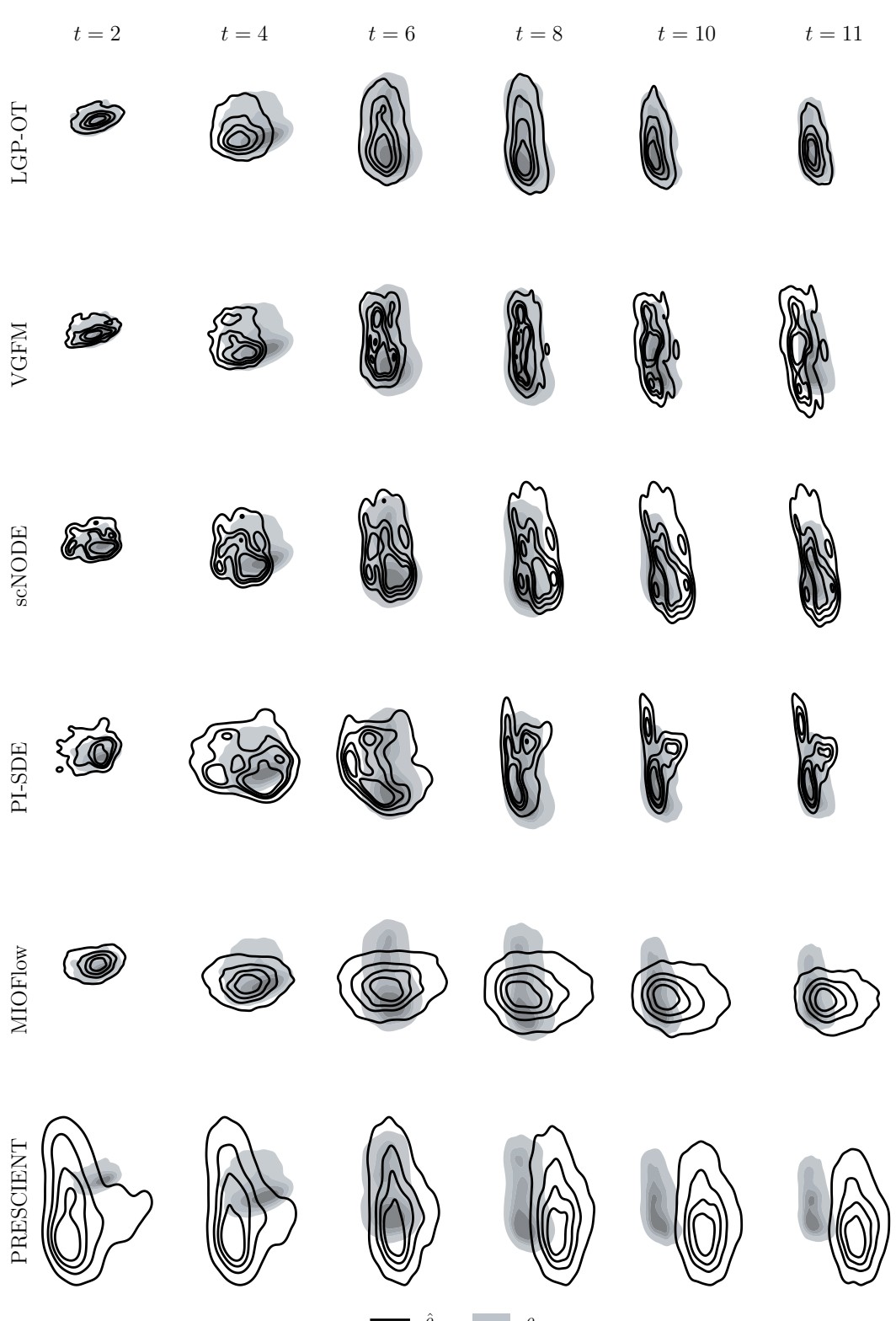

*Figure 10.* Comparison of predicted and observed distributions across models on test time points of the ZB hard task. The black contours, $\hat{\rho}$, represent the predicted cell density distributions generated by each dynamical model, while the grey shaded regions, $\rho$, depict the observed cell density.

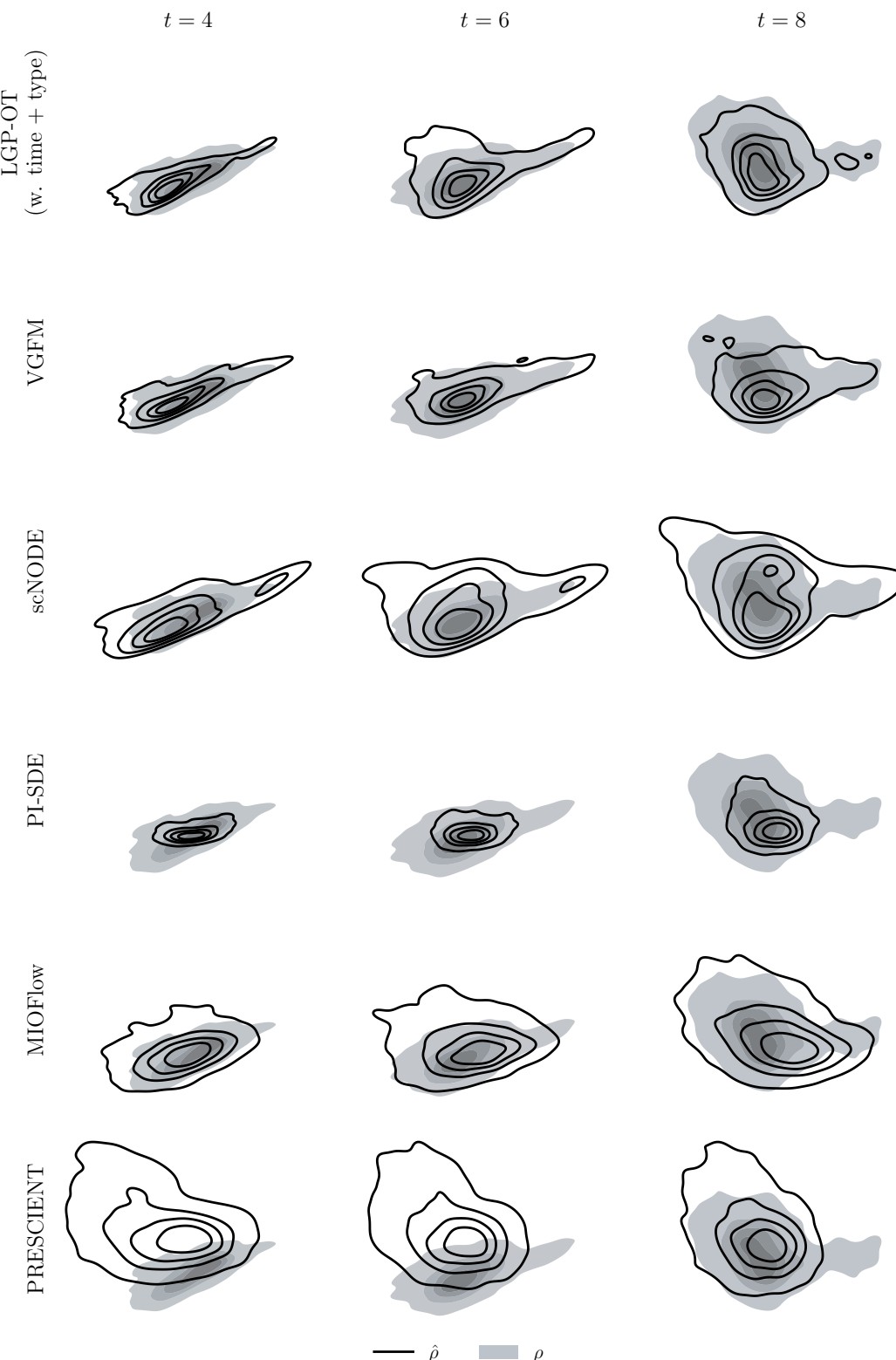

*Figure 11.* Comparison of predicted and observed distributions across models on test time points of the DR easy task. The black contours, $\hat{\rho}$, represent the predicted cell density distributions generated by each dynamical model, while the grey shaded regions, $\rho$, depict the observed cell density.

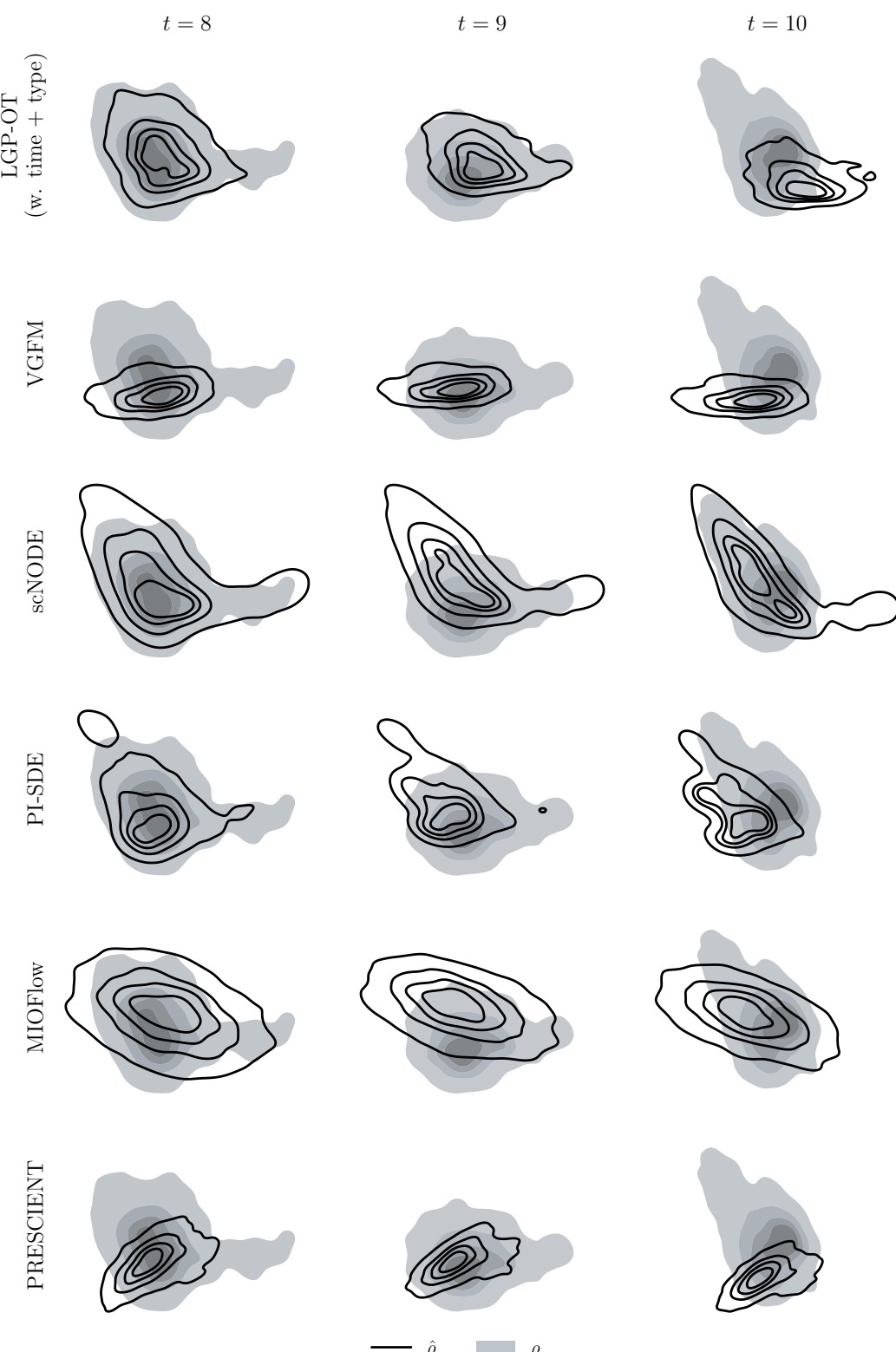

*Figure 12.* Comparison of predicted and observed distributions across models on test time points of the DR medium task. The black contours, $\hat{\rho}$, represent the predicted cell density distributions generated by each dynamical model, while the grey shaded regions, $\rho$, depict the observed cell density.

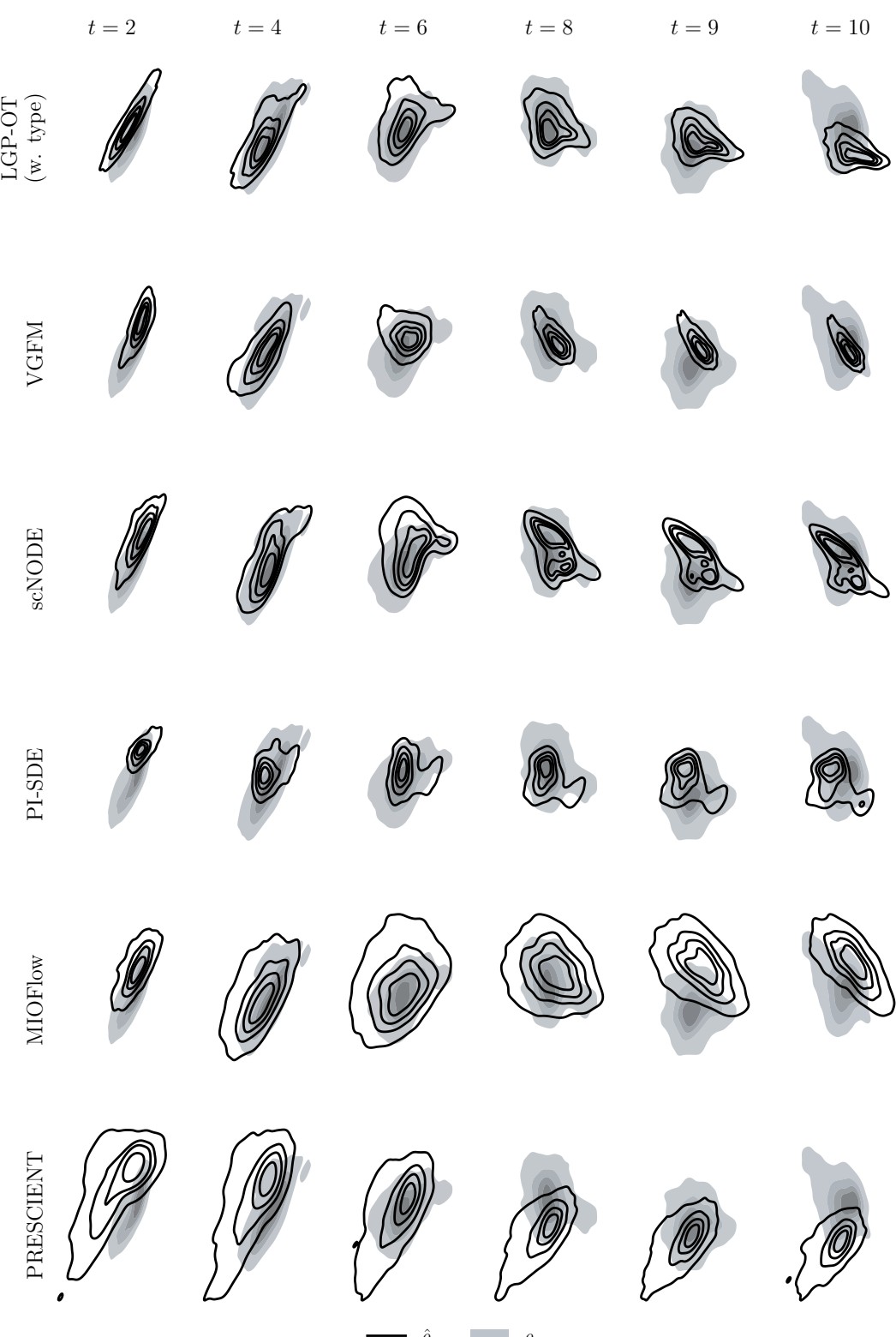

*Figure 13.* Comparison of predicted and observed distributions across models on test time points of the DR hard task. The black contours, $\hat{\rho}$, represent the predicted cell density distributions generated by each dynamical model, while the grey shaded regions, $\rho$, depict the observed cell density.

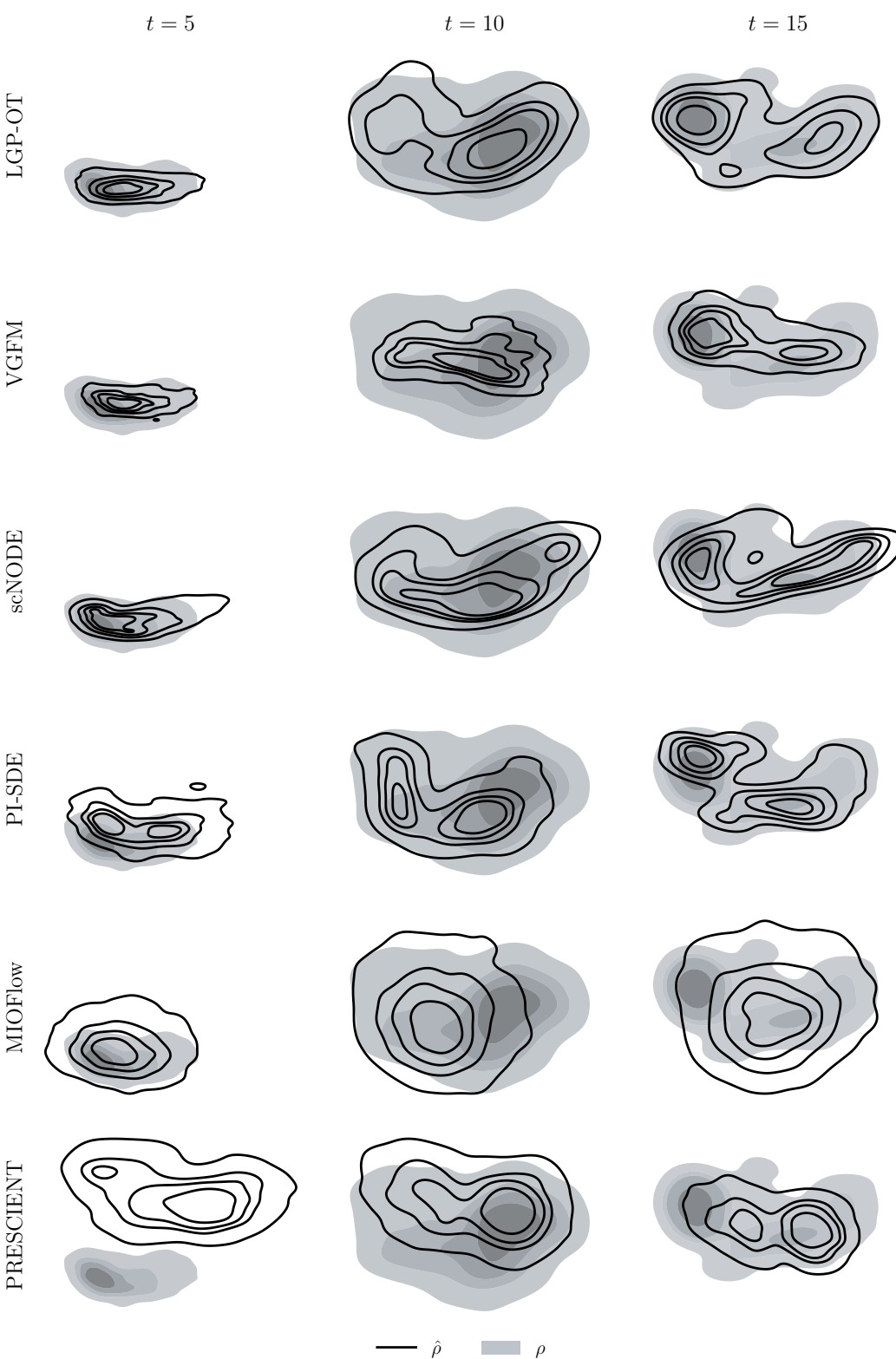

*Figure 14.* Comparison of predicted and observed distributions across models on test time points of the SC easy task. The black contours, $\hat{\rho}$, represent the predicted cell density distributions generated by each dynamical model, while the grey shaded regions, $\rho$, depict the observed cell density.

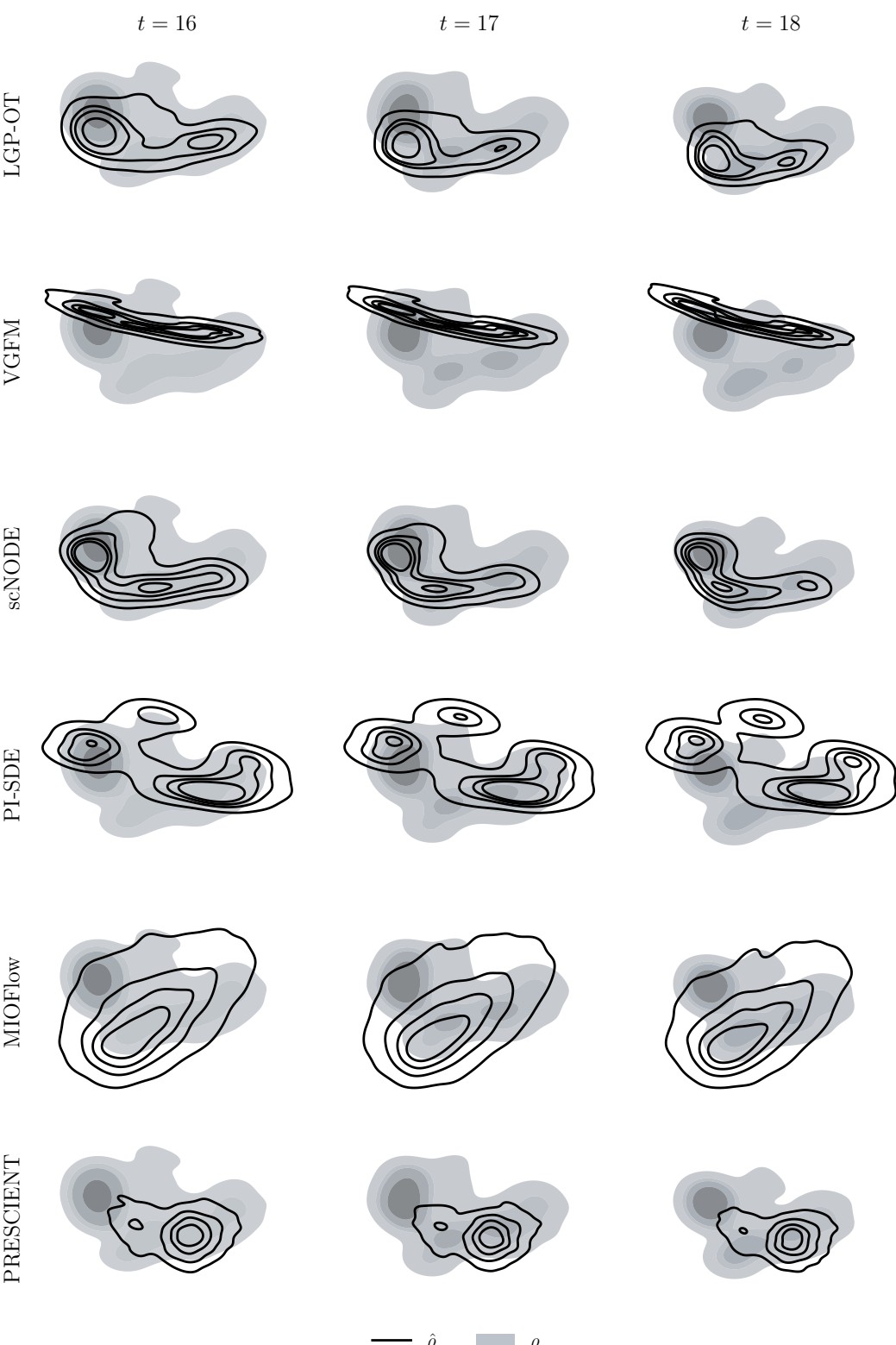

*Figure 15.* Comparison of predicted and observed distributions across models on test time points of the SC medium task. The black contours, $\hat{\rho}$, represent the predicted cell density distributions generated by each dynamical model, while the grey shaded regions, $\rho$, depict the observed cell density.

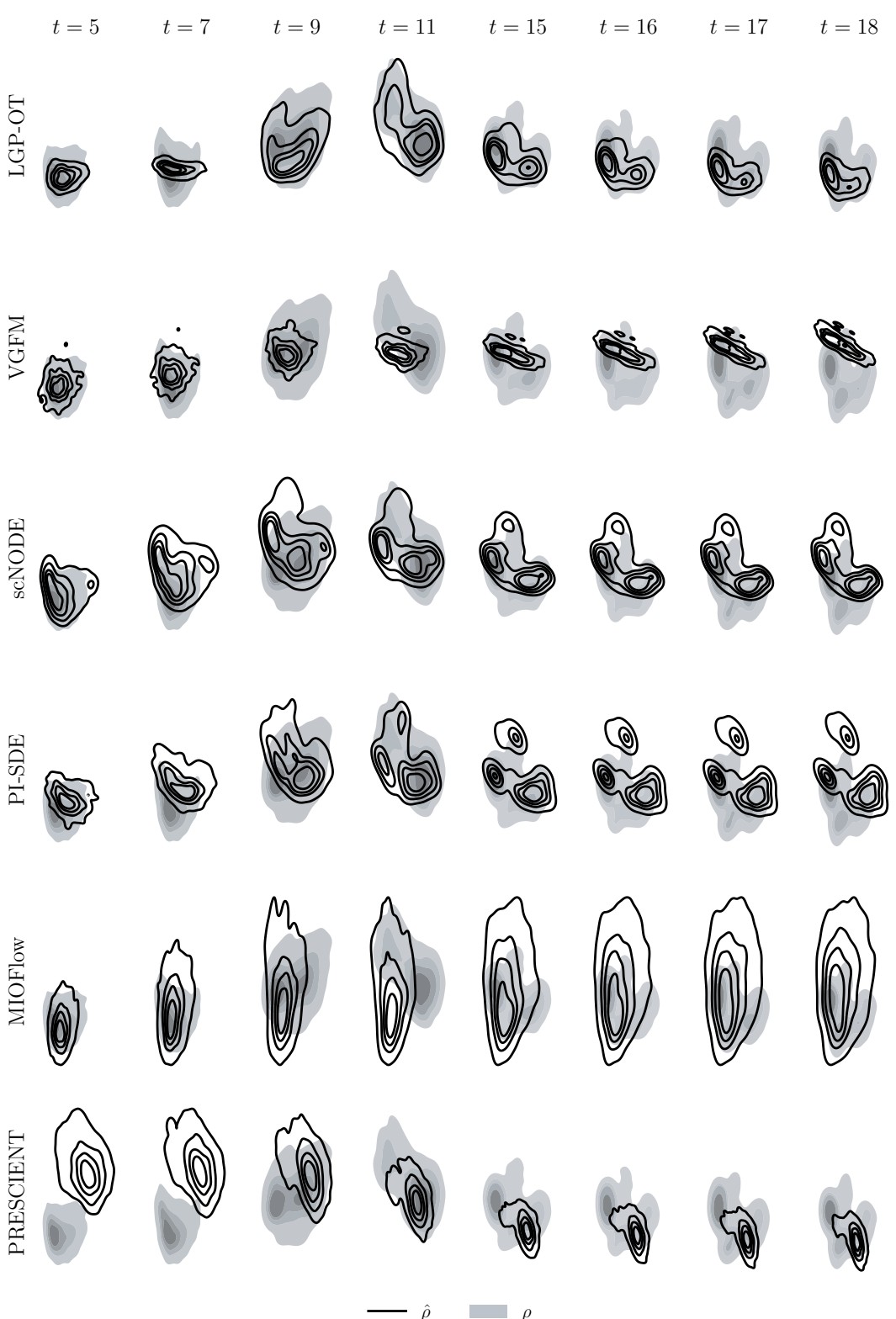

*Figure 16.* Comparison of predicted and observed distributions across models on test time points of the SC hard task. The black contours, $\hat{\rho}$, represent the predicted cell density distributions generated by each dynamical model, while the grey shaded regions, $\rho$, depict the observed cell density.

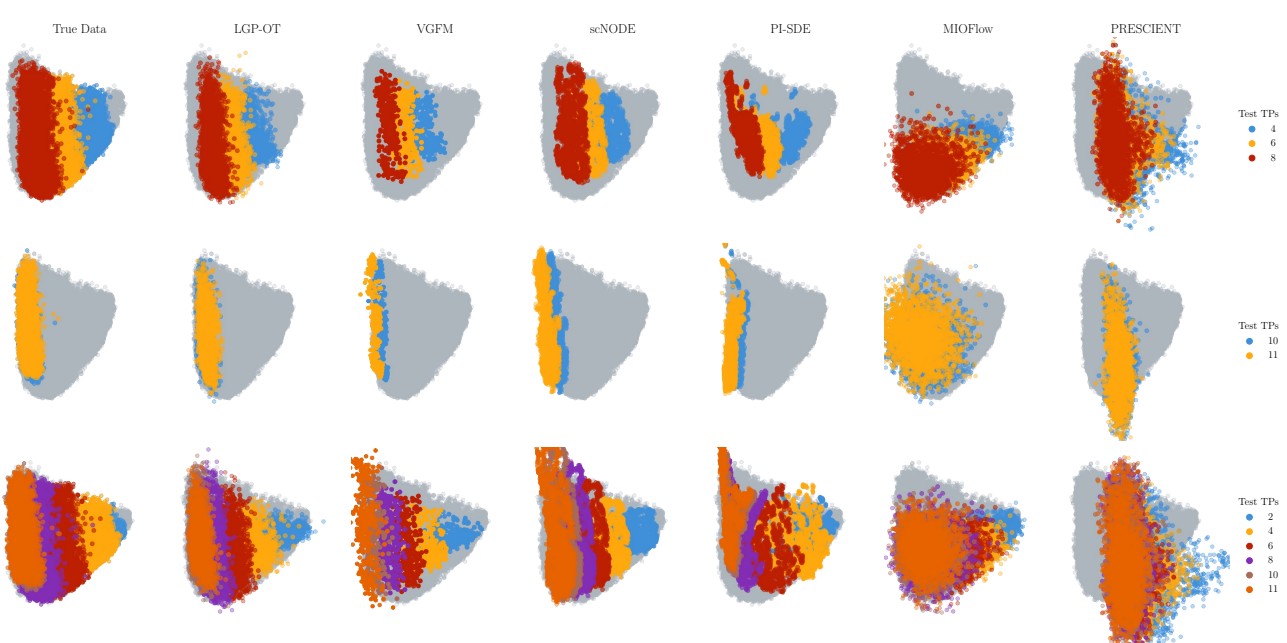

*Figure 17.* PCA visualization of the ground truth samples and model predictions on ZB data for easy (top row), medium (middle row), and hard (bottom row) tasks. The gray points represent the training data.

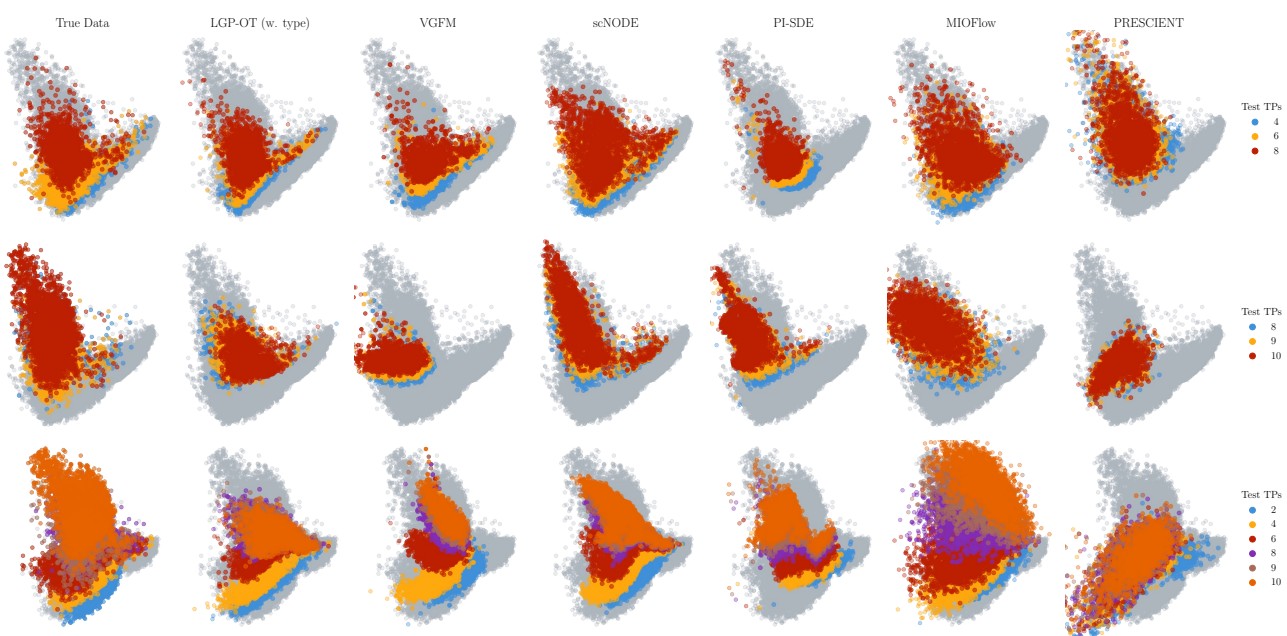

*Figure 18.* PCA visualization of the ground truth samples and model predictions on DR data for easy (top row), medium (middle row), and hard (bottom row) tasks. The gray points represent the training data.

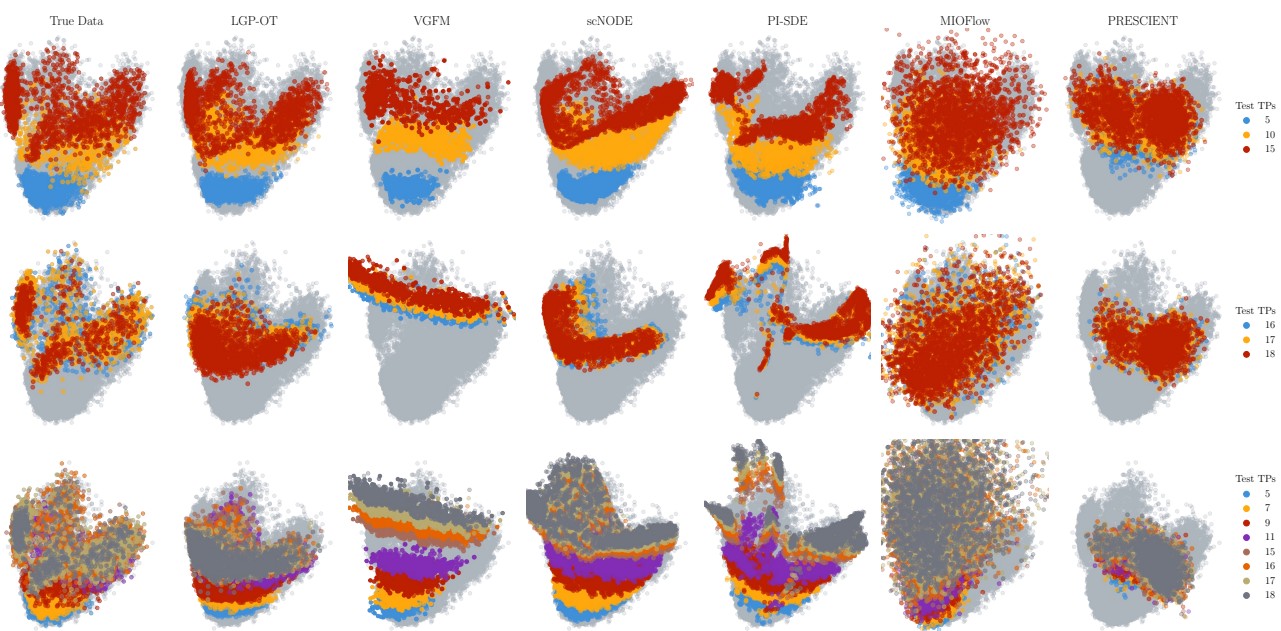

*Figure 19.* PCA visualization of the ground truth samples and model predictions on SC data for easy (top row), medium (middle row), and hard (bottom row) tasks. The gray points represent the training data.

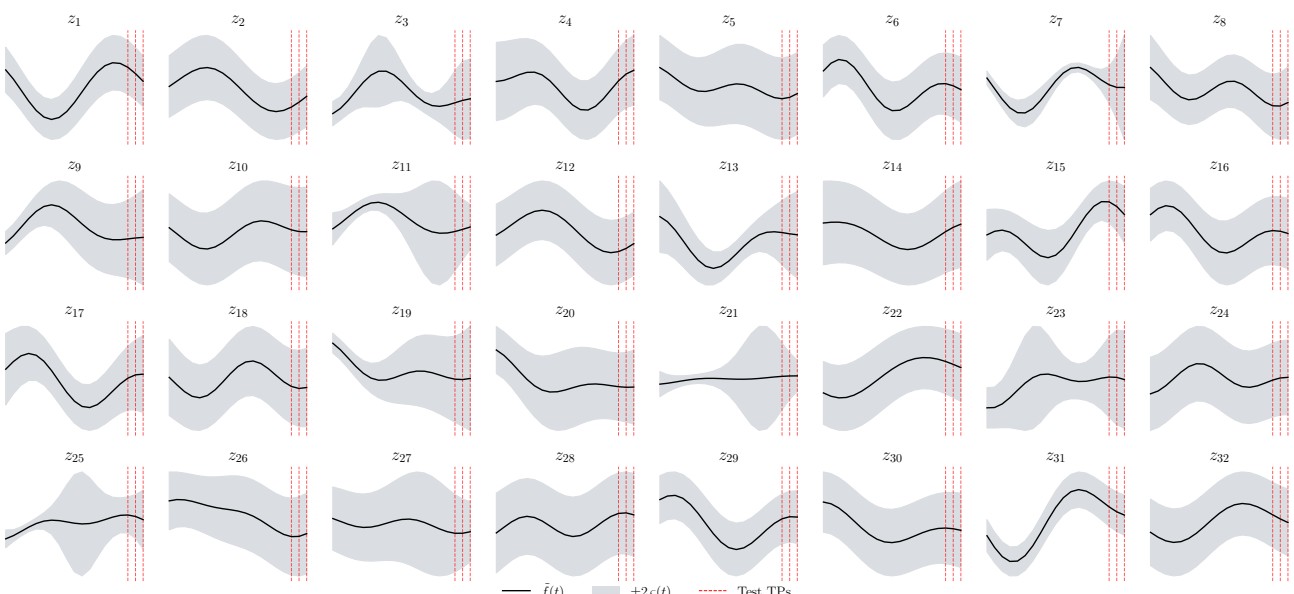

*Figure 20.* Learned latent temporal functions for the SC dataset under the medium task. The smooth trajectories and varying uncertainty bands illustrate the model's ability to capture both the population-level trends and the time-varying noise across latent dimensions.

