# OpenReview forum: "Modeling Temporal scRNA-seq Data with Latent Gaussian Process and Optimal Transport"
_ICML.cc/2026/Conference — ICML 2026 regular_

### Official Review · Reviewer_8Tzx · 2026-03-12

**Soundness:** 4
**Presentation:** 4
**Significance:** 3
**Originality:** 3
**Overall Recommendation:** 5
**Confidence:** 5

**Summary:**

The authors propose a VAE with a gaussian process (GP) prior for modeling the temporal dynamics of scRNAseq data. They make several modeling choices to improve predictive performance and computational efficiency. In particular, they allow for time-varying noise structure in the latent space to account for population heterogeneity during developmental inflection points; they also introduce categorical covariates to handle discrete data like cell types. Further, they statistically model cell measurement timings in order to capture temporal asynchrony across cells. The main efficiency gain comes from expressing the signal in the Laplacian eigenbasis, which means the resulting GP prior has a diagonal covariance structure and involves many fewer parameters than the otherwise dense NxN covariance matrix. The framework is trained with an OT loss between timepoints and is regularized to resemble their GP prior. The authors evaluate their model on temporal interpolation and extrapolation tasks, noting performance gains specifically due to cell-time variability and latent heteroscedasticity. Finally, using a perturbation technique which incorporates temporal dependencies, they show they can drive their simulated developmental processes towards target cell types.

**Compliance With Llm Reviewing Policy:**

Affirmed.

**Key Questions For Authors:**

*  LGP-OT learns a generative distribution over temporal trajectories, but could it be used to sample posterior developmental futures or simulate counterfactual dynamics beyond the observed time range? I.e. could you tell me, given a current state, how likely the gene expression distribution will look like X after a certain amount of time?

* I think the authors should address how the model would work in cases like cycling or abrupt branching. In the latter, maybe their asynchronous cell timing would have something interesting to say, since branches might rule out a single global time model for the whole population.

**Limitations:**

The authors do not discuss limitations. However, the questions I mention above hint at limitations. I.e. I am not sure how the model will be having in branching or cycling data.

**Strengths And Weaknesses:**

Soundness: The authors justify their particular architectural choices (a generative approach, varying noise strengths through time, explicitly temporal and cell-type modeling) both theoretically and in quantitative performance. Benchmarking against other models, ablation results, and performance on several data sets paint a convincing picture.

Presentation: Nicely written, easy to follow, clear figures, and respect for earlier work. Nice!

Significance: Modeling flows of scRNAseq data is a serious problem in computational biology, and the technical details explored here (varying noise levels, cell-time variability) are important and take steps in alleviating some of the inherent limitations of the underlying molecular measurement technology. However, I believe the main contribution–-the combination of latent modeling with temporal trajectories–could be explored more. For instance, LGP-OT learns a generative distribution over temporal trajectories, but could it be used to sample posterior developmental futures or simulate counterfactual dynamics beyond the observed time range? I.e. could you tell me, given a current state, how likely the gene expression distribution will look like X after a certain amount of time?

Also, I think the GP prior is a good assumption in many cases, but I think the authors should address how the model would work in cases like cycling or abrupt branching. In the latter, maybe their asynchronous cell timing would have something interesting to say, since branches might rule out a single global time model for the whole population.

Originality: They are clear that they build on existing methods, including those for latent space modeling of scRNAseq dynamics. However, the idea of regularization with a GP prior through time is, to my knowledge, a novel contribution. I also think the question of cell-time variability and categorical covariates are novel.

---

> ### Author Rebuttal · Authors · 2026-03-30
>
> We thank the reviewer for their interesting thought-provoking questions.
>
> **Simulating future distributions and counterfactual dynamics.** LGP-OT already provides population-level predictions beyond the observed time range, as demonstrated in our medium and hard extrapolation tasks (Tables 7–8). The GP prior naturally enables forecasting by evaluating the learned latent functions at arbitrary future time points and decoding the resulting latent states through $h_\theta$. In the revised manuscript, we will add some results for the extrapolation/future prediction task into the main text. For conditioning on a specific current cellular state, one could introduce an encoder (or approximate the inverse of $h_\theta$) to map an observed cell into the latent space. The inferred latent representation could be used to condition learned latent functions. This would yield a posterior over developmental futures conditioned on the cell's current gene expression profile. However, we did not implement cell-specific future predictions because those cannot be evaluated w.r.t. real data, due to the destructive nature of scRNA-seq technology.
>
> Our current framework does not explicitly model treatment or intervention variables, so counterfactual reasoning is not directly supported. However, our gradient-based perturbation strategy (Section 4.5) offers a step in this direction by modifying latent trajectories to satisfy gene expression constraints, enabling exploration of hypothetical scenarios. Extending LGP-OT with explicit treatment covariates is a promising direction for future work.
>
> **Branching.** The reviewer is absolutely right that biological cells do not commit to a lineage  simultaneously during a branching event, making a single global time model insufficient. Our current formulation is not designed to model branching events specifically but can handle multi-lineage experiments by conditioning trajectories on categorical cell types. Whereas, cell-specific latent time can disentangle individual cells from the global model by shifting a cell's position in time. Extending this framework to explicitly model branching (for instance, by integrating Branching GPs [1, 2] into latent space) is a natural and exciting direction.
>
> **Cycling.** Our current model relies on SE kernel for capturing smooth continuous latent dynamics. As a result, the current implementation cannot natively capture strict oscillatory or cyclic patterns. However, it is straightforward to use other kernels as long as they admit a similar basis function representation. Ruitort-Mayol et al. [3] showed that also periodic kernels admit a low rank approximation that has similar form as what we consider in our work. This adaptation would require no fundamental changes, making it straightforward to model oscillatory signals using periodic kernels within the LGP-OT framework.
>
> [1] Boukouvalas, A., Hensman, J., & Rattray, M. (2018). BGP: identifying gene-specific branching dynamics from single-cell data with a branching Gaussian process. Genome biology, 19(1), 65.
>
> [2] Sarkans, E., Ahmed, S., Rattray, M., & Boukouvalas, A. (2023). Modelling sequential branching dynamics with a multivariate branching Gaussian process. Transactions on Machine Learning Research.
>
> [3]  Riutort-Mayol, G., Bürkner, P. C., Andersen, M. R., Solin, A., \& Vehtari, A. (2023). Practical Hilbert space approximate Bayesian Gaussian processes for probabilistic programming. Statistics and Computing, 33(1), 17.

---

> > ### Author Rebuttal · Reviewer_8Tzx · 2026-04-03
> >
> > I thank the authors for their detailed response and for their commitment to adjust the manuscript in line with my suggestions. I think discussion for branching and cycling as areas for future development would be particularly interesting in the discussion/limitations section. I will keep my score as it is, representing my judgment that this is a strong paper that should be accepted.

---

### Official Review · Reviewer_fkrA · 2026-03-13

**Soundness:** 1
**Presentation:** 3
**Significance:** 3
**Originality:** 3
**Overall Recommendation:** 4
**Confidence:** 3

**Summary:**

The paper introduces a framework based on latent heteroscedastic GPs, approximated via Hilbert space methods for scalability, for temporal single cell transcriptomic data prediction. The framework models cell specific latent time and allows for cell type conditioning via a product kernel to capture cell type covariates. Together, these components of the framework allow it to capture temporal asynchrony and multi-lineaage structure. Within this framework, the paper also introduces a gradient-based method to simulate gene perturbations across the temporal domain.

**Compliance With Llm Reviewing Policy:**

Affirmed.

**Final Justification:**

Originally my score was 2 and my confidence level was 4, with my main concerns being the lack of an encoder in this temporal single cell transcriptomic data prediction framework. After the authors' rebuttal and subsequent discussion, this has alleviated my concern somewhat on this front. Accordingly, I raise my score to a 4, but lower my confidence to a 3. I believe the discussion on the pros and cons of having an encoder in temporal scRNA-seq prediction is valuable and should be expanded upon in the paper.

**Key Questions For Authors:**

1. Does the framework utilize an neural network encoder to learn the latent space? If so how is it trained? Note that in Figure 1 and the generative process described in section 4.2, an encoder is conspicously missing. However section 3.3 discusses extending the standard VAE framework by replacing the standard normal prior in the latent space with a GP and lines 276-279 in the right column note that “LGP-OT and scNODE train a non-linear mapping between the high-dimensional observation space and latent space simultaneously with training the dynamics model in the latent space”. If the mean field variational distribution $q_\Psi(\Phi)$ described in lines 225-226 is realized in the form of a neural network encoder with weights parameterized by $\Psi$, it would be good to be clear with that.
2. The main table of results (Table 1) follow from the experimental setting from [1]. However the corresponding table in [1] (Table 2) report very different numbers than the results in Table 1 of this paper. What would explain this discrepancy, especially for the methods benchmarked in both tables?
3. How is the cell-specific time variability component different from pseudotime inference methods that utilize a VAE framework, such as [2]? If the cell-specific time variability component is indeed performing a form of pseudotime inference, it would be good to make and expand upon this connection.
4. What exactly is Figure 2 showing? Since the gene expression dimensionality is 2000 (Section C.1) there must be some form of dimensionality reduction performed to obtain Figure 2. It would be good clarify the steps used to generate the figure.

I would be willing to raise my score if the authors could address the issues raised above.

[1] scNODE: generative model for temporal single cell transcriptomic data prediction - https://academic.oup.com/bioinformatics/article/40/Supplement_2/ii146/7749071

[2] scTour: a deep learning architecture for robust inference and accurate prediction of cellular dynamics - https://link.springer.com/article/10.1186/s13059-023-02988-9

**Limitations:**

Yes

**Strengths And Weaknesses:**

Strengths:
- The paper introduces a novel framework in using a latent GP in order to model the temporal dynamics of cell trajectories.
- The use of Hilbert space methods for GP inference is unique and certainly helps with scalability.

Weaknesses:
- See questions below.

---

> ### Author Rebuttal · Authors · 2026-03-30
>
> We thank the reviewer for their questions.
>
> **Response to Q1.** LGP-OT does not use a neural network encoder. Unlike standard VAEs, our framework directly parameterizes the global latent trajectory through the Hilbert space basis function expansion $\boldsymbol{z}(t) = \boldsymbol{A\phi}(t)$, $\boldsymbol{\phi}$ are fixed basis functions (Eq. 4) and $\boldsymbol{A} \in \mathbb{R}^{L \times M}$ whose prior depends on the GP kernel hyperparameters via the spectral density. The mean-field variational distribution $q_\Psi(\Phi)$ directly parameterizes the means and variances of the distributions over the weight matrices $\boldsymbol{A}$ and kernel hyperparameters, which are optimized via stochastic gradient descent using the reparameterization trick.
>
> We acknowledge that the reference statements may create ambiguity. The "non-linear mapping" refers exclusively to the decoder (push-forward) network $h_\theta: \mathcal{Z} \rightarrow \mathcal{X}$, which is trained jointly with the variational parameters by minimizing the OT objective (Eq. 7). This is why Figure 1 intentionally omits an encoder: the model is a decoder-only generative framework with a GP-structured latent space whose parameters are learned variationally. We will revise the manuscript to make this distinction clearer.
>
> **Response to Q2.** The authors of the scNODE paper report the unbiased Sinkhorn divergence estimates provided by the GeomLoss package [1] that uses the square of the distance metric $d(x, \hat{x}) = 0.5 \lVert x- \hat{x} \rVert_2^2$ and also includes the entropic regularization term. Whereas in our paper, we report the exact Wasserstein distance with distance metric $\lVert x- \hat{x} \rVert_2$ without any entropic regularization term, the same as other baselines report in their original publications. For comparison, below we present the Sinkhorn divergence comparisons like in the scNODE paper for DR/Hard:
>
> |Method\t|2|4|6|8|9|10|
> |-|-|-|-|-|-|-|
> |LGP-OT|429.8|465.6|519.9|567.2|569.1|722.6|
> |LGP-OT (w. type)|411.0|459.3|515.5|565.7|558.1|709.4|
> LGP-OT (w. time)|435.3|462.2|519.1|569.3|563.5|705.9|
> LGP-OT (w. time + type)|413.2|461.5|515.9|566.6|558.1|710.9|
> VGFM|455.3|487.3|553.3|630.4|621.7|737.6|
> scNODE|445.0|466.3|534.8|598.3|594.0|722.6|
> MIOFlow|439.8|477.2|536.3|616.1|672.0|855.5|
> PI-SDE|457.2|471.4|529.3|572.8|581.9|740.4|
> PRESCIENT|530.5|519.8|542.7|624.8|578.2|723.4|
>
> **Response to Q3.** The key distinction lies in the temporal reference frame. Classical pseudotime methods like the referenced model are typically designed for settings where temporal annotations are absent. They infer a virtual ordering of cells along a developmental axis, typically from a single snapshot. In contrast, assuming time-series scRNA-seq data, the latent time $\tau$ in our model is anchored to the actual measurement time $t$ via an informative prior, capturing small deviations from the measurement time rather than constructing a de novo ordering. This reflects the biological reality that cells within the same snapshot may differ slightly (temporal asynchrony), while remaining grounded in the experimentally recorded time. Thus, while both approaches address temporal heterogeneity, pseudotime methods recover a latent ordering without real-time information, whereas our formulation refines known time points to accommodate within-snapshot variability. We will clarify the distinction between pseudotime and actual time-series modeling in the revised manuscript. To our knowledge, Reid & Wernisch [2] and Lönnberg et al. [3] were the first to propose using this model for cell-specific asynchrony, and we cite them in our manuscript.
>
> **Response to Q4.** Thank you for pointing this out. Figure 2 visualizes the predicted and observed cell distributions projected onto the first two principal components. We applied PCA to reduce the 2000-dimensional gene expression space to two dimensions and used kernel density estimation to plot the contours. We will add this clarification to the figure caption in the revised manuscript.
>
> [1] https://www.kernel-operations.io/geomloss/api/pytorch-api.html
>
> [2] Reid, J. E. and Wernisch, L. Pseudotime estimation: deconfounding single cell time series. Bioinformatics, 32(19): 2973–2980, 2016.
>
> [3] Lonnberg, T., et al. Single-cell rna-seq and computational analysis using temporal mixture modeling resolves th1/tfh fate bifurcation in malaria. Science immunology, 2(9):eaal2192, 2017.

---

> > ### Author Rebuttal · Reviewer_fkrA · 2026-04-03
> >
> > I thank the authors for their response. I'm satisfied with their responses to Q2-4.
> >
> > Follow up on their response to Q1, what is the reasoning for omitting an encoder? Specifically, I'm concerned about the viability of the framework without cell-specific gene expression information from an encoder informing the representation of the latent variables, as done in scNODE and other similar VAE-based works.

---

> > > ### Author Response · Authors · 2026-04-06
> > >
> > > Thank you for this insightful follow-up. The omission of an encoder is a deliberate design choice motivated by the fundamental nature of scRNA-seq data, and we believe it is not only viable but advantageous for our setting.
> > >
> > > **The data only inform population-level dynamics.** Latent variable models and VAEs have become very popular for analyzing scRNA-seq data, which naturally leads to thinking in terms of embeddings for individual cells. However, due to the destructive nature of scRNA-seq, the data neither allow us to train nor evaluate models that predict individual cell trajectories. The data only tells us how the marginal distribution of cells develops over time. While latent variable models can produce per-cell latent codes using either amortized variational inference (VI) (encoder-decoder structure) or non-amortized VI (decoder model only; we elaborate on this below), the temporal predictions of individual cell trajectories cannot be evaluated against ground truth. Otherwise, standard ELBO-based approaches would suffice for training. Importantly, this limitation applies equally to encoder-based models: although scNODE has a pre-trained VAE encoder, its dynamics model is trained and evaluated using only the OT loss between population distributions, not on predicting individual cell trajectories. In other words, the data allow us to train and assess only how the distribution of cells develops over time, which is precisely what LGP-OT targets directly.
> > >
> > > **Our design reflects this.** LGP-OT models population-level temporal dynamics through GP-parameterized weight matrices $\boldsymbol{A}$, which define how the latent distribution evolves. Individual cells are generated by sampling in the latent space and passing through the heteroscedastic latent process (Eq. 6), followed by the decoder. Gene expression information from the observed cells is incorporated through the OT objective: the gradient of the Wasserstein distance with respect to the variational parameters $\Psi$ propagates information from observed cells back into the latent trajectory parameters. Thus, the observed gene expression of cells shapes the latent representation, but through a distributional matching mechanism rather than per-cell encoding.
> > >
> > > Encoder-based approaches like scNODE couple two functions into the encoder: learning a low-dimensional representation and providing initial conditions for the dynamical system to solve the initial value problem over time. While an encoder is useful for inspecting individual cells in latent space, it is unnecessary for the population-level modeling goal and introduces challenges. The authors of scNODE themselves identify that a fixed encoder trained on observed time points may not generalize well to unobserved time points with shifted distributions, which is precisely why they introduce dynamic regularization to iteratively update the encoder (Section 2.3 of the scNODE paper) using the ODE as the prior. Our approach avoids this issue by construction. Because the GP prior constrains the latent trajectory to vary smoothly, the decoder encounters latent states at unobserved time points that remain within or close to its training distribution.
> > >
> > > **Per-cell embeddings are still possible.** Single-cell latent embeddings can also be obtained in decoder-only models by carrying out non-amortized inference. In our framework, this corresponds to inverting the push-forward mapping such that $\hat{z}_n=\mathrm{argmin}\ d(x_n,h(z_n))$, where $d$ is the distance metric used in Eq 1. Cell-specific trajectory could then be predicted by conditioning our latent heteroscedastic GP model at the obtained embedding $\hat{z}_n$ to predict other time points. However, as discussed, the predictive performance of such per-cell trajectory models, whether from our framework or from encoder-based methods, cannot be evaluated from snapshot data. But single-cell-specific embeddings can be obtained both in encoder-decoder and decoder-only model structures if they are deemed important for downstream analysis tasks.
> > >
> > > **Empirical evidence.** Our perturbation analysis provides direct evidence that the learned latent space captures relevant structure. Perturbing DE genes steers the generated population toward the expected cell types (e.g., 93% Hindbrain or 92% PSM classification accuracy under cell type-specific guidance), while perturbing non-DE genes produces no significant shift. This would not be possible if the latent space lacked relevant cell-level organization. Furthermore, LGP-OT's extrapolation performance confirms that the encoder-free design does not sacrifice representational quality. Furthermore, on the SC/Hard task (Table 8):
> > >
> > > ||15|16|17|18|
> > > |-|-|-|-|-|
> > > LGP-OT|14.97|15.46|15.44|16.29|
> > > scNODE|16.76|17.90|17.26|18.14|
> > >
> > > The gap at later time points suggests that our approach generalizes more robustly to unobserved time points than the encoder-based alternative.
> > >
> > > We will clarify this reasoning in the revised manuscript.

---

### Official Review · Reviewer_TVy4 · 2026-03-13

**Soundness:** 3
**Presentation:** 3
**Significance:** 3
**Originality:** 3
**Overall Recommendation:** 4
**Confidence:** 3

**Summary:**

The paper proposes LGP-OT, which combines a latent heteroscedastic Gaussian process prior, optimal transport matching in observation space, optional cell-specific latent time and cell-type conditioning, and a gradient-based perturbation procedure for temporal trajectories. The study's important contribution pertains to replacing neural ODE/SDE-style latent dynamics with a GP-based population model that appears empirically strong on interpolation/extrapolation benchmarks over three public datasets.

**Compliance With Llm Reviewing Policy:**

Affirmed.

**Final Justification:**

addressed main concerns

**Key Questions For Authors:**

- Objective and probabilistic interpretation.

In Appendix C, the final training loss is obtained by replacing the element-wise log-likelihood term in the ELBO with an OT distance. Could the authors clarify whether this objective corresponds to a principled variational bound under a well-defined generative model, or whether it should instead be viewed as a heuristic surrogate objective? If it is the latter, what guarantees or intuitions justify this replacement?

- Ablation of each component.

The method combines several ingredients: heteroscedastic latent GP, OT matching, latent cell-time modeling, and cell-type conditioning. Could the authors provide a cleaner ablation table quantifying the marginal contribution of each component across datasets and task types, rather than mainly reporting selected variants? This would make it easier to assess which parts are truly essential.

- Fairness of hyperparameter tuning across baselines.

The paper states that all models were tuned to their optimal hyperparameters. Could the authors provide the exact tuning budget and protocol for each baseline, including search space size, number of trials, validation criterion, and whether the same computational budget was used across methods? This is important because the reported gains over the strongest baselines are sometimes not very large.

**Limitations:**

yes

**Strengths And Weaknesses:**

## Strengths

1. **Well-motivated and fairly complete method design.** The paper integrates a heteroscedastic latent GP, OT-based distribution matching, cell-specific latent time, and cell-type conditioning into a unified framework, which is well aligned with the core challenge of snapshot scRNA-seq data where true single-cell temporal correspondences are unavailable.

2. **Relatively comprehensive experimental evaluation.** The authors evaluate on three public datasets (ZB, DR, and SC) under easy / medium / hard interpolation and extrapolation settings, and compare against strong baselines such as scNODE, MIOFlow, PRESCIENT, PI-SDE, and VGFM. The method shows consistently competitive and often leading performance, especially on harder tasks.

3. **Beyond quantitative results, the paper also includes qualitative visualizations and sensitivity analyses.** In addition, the perturbation analysis suggests that the model is intended not only for prediction but also for generating biologically meaningful hypotheses.

## Weaknesses

1. **The novelty mainly comes from the integration of existing components rather than a fundamentally new technical idea.** The GP prior, Hilbert-space approximation, OT matching, latent time, and categorical kernel are each not new by themselves. The main contribution lies in combining them for temporal scRNA-seq modeling, but the paper does not yet provide a sufficiently deep explanation of why this particular combination should outperform ODE/SDE/flow-based approaches.

2. **The theoretical motivation of the objective is not fully rigorous.** In the appendix, the derivation appears to begin with a VAE likelihood under an assumed one-to-one correspondence, and then replaces the reconstruction term with an OT loss. This feels more heuristic than a fully principled probabilistic derivation, and the connection between the final objective and likelihood, generalization, or identifiability remains somewhat unclear.

---

> ### Author Rebuttal · Authors · 2026-03-30
>
> We thank the reviewer for their comments.
>
> **Response to W1.** While ODE/SDE methods are well-motivated, they integrate dynamics recursively, which increases runtime as serial execution cannot be parallelized, makes training more challenging, especially for longer trajectories due to an increasingly complex loss landscape [1], and can cause error accumulation that degrades extrapolation. Our GP defines a global function over time: predictions at any time point are obtained by direct basis function evaluation, not sequential integration. The kernel's smoothness prior provides principled regularization at unobserved times, explaining our robust performance in hard tasks where baselines degrade. Additionally, biological variability changes across developmental stages. Our heteroscedastic GP captures time-varying noise naturally, and consistent gains over homoscedastic ablations (Tables 7, 8) confirm this is not redundant.
>
> **Response to W2 and Q1.** We acknowledge that our final objective does not correspond to a variational bound under a single unified generative model, and the ELBO derivation in Appendix B is not intended to claim that the full objective is a tight variational bound. Instead, Appendix B serves a precise technical purpose: to derive the KL regularization terms that enforce the smoothness constraints imposed by the GP prior on the Hilbert space basis function parameterization. These terms regularize the distributions over basis function weights and kernel hyperparameters, and their form follows rigorously from the GP prior structure.
>
> Having established these regularization terms, we utilize an OT-based objective, which is a standard practice in the field. All state-of-the-art methods, including the methods we compare against, rely on loss functions that consist of 1) OT or related distribution-matching objective, and 2) a regularization term specific to their modeling approach (e.g., dynamic latent regularization in scNODE, Hamiltonian-Jacobi regularization in PI-SDE). Our objective follows the same paradigm: an OT loss for distribution matching plus principled regularization derived from the GP prior.
>
> The justification is twofold. First, the OT cost is computed in expectation over the variational distribution of the GP parameters (Eq. 7), so the transport cost remains internally consistent with the uncertainty and smoothness of the latent GP structure. Second, this objective can be understood through the lens of the Wasserstein autoencoder framework [2], which establishes that minimizing a transport cost in observation space combined with a latent regularization term constitutes a valid approach to generative modeling. We discuss this connection in Section 4.2. We will clarify our motivation and objective function in the revised manuscript accordingly.
>
> **Response to Q2.** We kindly refer the reviewer to Tables 7 and 8 in the Appendix, which provide all ablations for all datasets and tasks. We note that results for the cell-type conditioning are only available for the DR dataset, as other datasets do not contain cell-type annotations for the entire trajectory, as explained in Section 5, 305-312. We also note that OT based loss is not an optional ingredient that we can use instead of likelihood-based loss, but a necessity, as explained above. We will provide a cleaner and separate table in the main text for the ablation results. We're happy to elaborate more if the reviewer finds the current ablation results insufficient and specifies what is insufficient.
>
> **Response to Q3.** Full details are provided in Appendix C2. All models were tuned using 3-fold cross-validation on training data with $W_2$ distance as the validation criterion. For scNODE, MIOFlow, and PRESCIENT, we adopted the optimal hyperparameter configurations established through grid search that was already conducted in [3], which used the same datasets and evaluation protocol. For PI-SDE and VGFM, we followed the hyperparameter tuning strategies described in their respective publications. For LGP-OT, we performed max. 45 trials over the space detailed in Appendix C2. We did not impose an identical fixed computational budget across methods. All models were trained until convergence using their recommended optimization protocols.
>
> [1] Ribeiro, A. H., Tiels, K., Umenberger, J., Schön, T. B., \& Aguirre, L. A. (2020). On the smoothness of nonlinear system identification. Automatica, 121, 109158.
>
> [2] Tolstikhin, I., Bousquet, O., Gelly, S., & Schoelkopf, B. (2018, February). Wasserstein Auto-Encoders. In International Conference on Learning Representations.
>
> [3] Zhang, J., Larschan, E., Bigness, J., & Singh, R. (2024). scNODE: generative model for temporal single cell transcriptomic data prediction. Bioinformatics, 40(Supplement_2), ii146-ii154.

---

> > ### Author Rebuttal · Reviewer_TVy4 · 2026-04-04
> >
> > Address my concern

---

### Official Review · Reviewer_N985 · 2026-03-15

**Soundness:** 3
**Presentation:** 4
**Significance:** 3
**Originality:** 3
**Overall Recommendation:** 4
**Confidence:** 3

**Summary:**

The authors propose a new method for modeling temporal scRNA-seq data. It is based on learning a latent Gaussian process that is decoded to predict the distribution of cells at different points in time. The reconstruction loss is replaced with a distribution-based loss. The authors provide strong empirical results and comparisons against multiple methods.

**Compliance With Llm Reviewing Policy:**

Affirmed.

**Key Questions For Authors:**

p3 l161: Could add a citation for the Sinkhorn algorithm.

p4 l167: What is a stationary kernel in this context?

p4 l210: Is the gene expression space also normalized?

p5 l235: It looks like the OT loss term could potentially be replaced with another distribution loss, such as MMD. Could the authors comment on this and whether they tried other losses?

p6 l281: For simulating gene perturbations in the latent space: do you perturb directly in the latent space or in the ambient space? Since the latent space is smaller, how do you identify which dimensions to modify?

**Limitations:**

The authors should add a discussion of the limitations of the method, or highlight them through additional experiments.

**Strengths And Weaknesses:**

### Strengths

Cell-specific latent time is a great advantage that many methods do not provide. However, it would be great to have more experiments highlighting this capability.

The introduction and related work are great. In particular, the related work section clearly highlights the shortcomings of previous methods. The experiments seems rigorous: the authors clearly explain data processing and the hyperparameter search for each model. The comparisons are of high quality and include models learning different types of dynamics. Overall, the paper seems well polished both in terms of theory and experimentation.

### Weakness

The authors should add a visualization or an experiment on the latent time, perhaps comparing it with diffusion time.

Replacing the MSE-type loss (reconstruction) with a distribution loss appears more heuristic. The authors should provide a toy example where pairing is available and compare the effect of using either MSE or OT loss. This would provide a proof of concept that replacing this loss is reasonable.

I may have missed it, but there is no comment or experiment on the training time or computational complexity of the model. Given how high-dimensional the data of interest can be, it would be helpful for the reader to have an idea of the scaling and complexity of the model.

---

> ### Author Rebuttal · Authors · 2026-03-30
>
> We thank the reviewer for their comments.
>
> **Response to W1.** Here we anonymously present the learned latent times of cells on the DR/medium: https://ibb.co/27NGTq33. Each dot represents a single cell, with the x-axis position denoting the observed measurement time point and the y-axis denoting the estimated latent time. We will add these visualizations to the Appendix.
>
> **Response to W2.** We would like to clarify the motivation behind our objective function. The use of OT-based losses for learning dynamics from unpaired data observed as snapshots of the population is well-established in this literature. Indeed, all state-of-the-art methods, including the methods we compare against, rely on loss functions that consist of 1) OT or related distribution-matching objective, and 2) a regularization term specific to their modeling approach (e.g., dynamic latent regularization in scNODE, Hamiltonian-Jacobi regularization in PI-SDE). OT-type losses are used precisely because the destructive nature of scRNA-seq precludes cell-level pairing. Our contribution is not the replacement of MSE with OT per se, but rather the principled integration of a GP-regularized latent space with the OT objective. Concretely, the ELBO derivation in Appendix B serves a specific technical purpose: it establishes the correct KL regularization terms that enforce the smoothness properties implied by the GP prior when the model is parameterized via the Hilbert space basis function approximation. These KL terms regularize the distributions over basis function weights and kernel hyperparameters, ensuring that the learned latent trajectories respect the temporal smoothness encoded by the GP. We follow the literature and use OT loss for unpaired data while regularizing the latent temporal model with a GP prior. Incorporating the uncertainty of the latent GP model also into the OT loss via expectation ensures that the transport cost is internally consistent with the GP-imposed smoothness structure. We will clarify these aspects in the revised manuscript.
>
> **Response to W3.** LGP-OT's scalability stems from the Hilbert space GP approximation, which reduces the GP training cost from $O(N^3)$ to $O(NM)$ per latent dimension, and entropic regularization of the OT objective enabling approximate linear-time Sinkhorn computation. Unlike neural ODE methods that require sequential integration, LGP-OT evaluates trajectories simultaneously via closed-form basis functions. We provide training time comparison against the most competitive baselines on the largest dataset (SC):
>
> ||# Params|Training Time (s)|Inference Time (s)|
> |-|-|-|-|
> |LGP-OT|105K|1609|0.25|
> |scNODE|217K|1082|0.36|
> |VGFM|434K|961|8.45|
> |PISDE|182K|9617|3.85|
>
> LGP-OT achieves competitive training time with the fewest parameters among all methods, while offering a fast inference. The modest training overhead relative to scNODE and VGFM is offset by LGP-OT's substantially better predictive performance.
>
> **Responses to questions.**
>
> - The reference for Sinkhorn is already in the paper p3 l159 [1].
>
> - A stationary kernel is a covariance function that depends only on the difference between inputs, $k(t, t') = k(|t -t'|)$, rather than their absolute values. While this assumes a translation-invariant correlation structure, our model captures non-stationary behavior through the heteroscedastic noise model, which allows variance to change over time. Exploring non-stationary kernels remains an interesting direction for future work. Yet, Hilbert space approximation is not applicable for non-stationary kernels.
>
> - Eqs. 11 and 12 in Appendix C1 present the normalization and log-transformation of raw gene counts.
>
> - It is possible to utilize any distribution-matching-based loss. In our experiments, we followed the literature as Wasserstein is a standard approach. For demonstration purposes, here we provide MMD comparison of LGP-OT and scNODE when they are trained and evaluated using MMD (Gaussian kernel with multiscale median heuristic) instead of OT on SC/Hard:
>
> ||5|7|9|11|15|16|17|18|
> |-|-|-|-|-|-|-|-|-|
> LGP-OT|0.15|0.11|0.08|0.05|0.05|0.09|0.11|0.21|
> scNODE|0.19|0.12|0.10|0.13|0.07|0.10|0.12|0.20|
>
> - Perturbations are defined in the gene expression space by specifying target genes and their scaling factors. The variational parameters governing the latent trajectories are then updated by backpropagating gradients of the OT cost through the decoder, where the OT cost is computed w.r.t. the perturbed gene expression profiles. This means we never need to identify specific latent dimensions. The differentiable (fixed) decoder automatically translates gene-level perturbation targets into the appropriate latent space adjustments. We refer to [2] who proposed a similar approach, although without the temporal dimension.
>
> [1] Cuturi, M. Sinkhorn distances: Lightspeed computation of optimal transport. Advances in neural information processing systems, 2013
>
> [2] https://www.mdpi.com/2073-4425/16/12/1439

---

> > ### Author Rebuttal · Reviewer_N985 · 2026-04-05
> >
> > Regarding W2, I understand that other methods also employ optimal transport losses; however, they do not necessarily introduce a new ELBO. In Section B2, the authors state:
> >
> > > By substituting the log-likelihood term in Equation (8) with the
> > regularized OT cost, and minimizing the negative ELBO, we arrive at the final objective function presented in Equation (7):
> >
> > The authors should justify that this substitution is sound, as it forms the basis of the method’s definition. A simple experiment on a toy dataset could help support this claim.
> >
> > Regarding citation [1], the paper introduces the entropy-regularized optimal transport formulation, which is distinct from the Sinkhorn–Knopp algorithm used to solve it.
> >
> > Thank you for the additional experiments.

---

> > > ### Author Response · Authors · 2026-04-07
> > >
> > > We thank the reviewer for this valuable observation. We acknowledge that the sentence referred to (l 773-774) is imprecisely worded and misleading. We emphasize that the final objective is not an ELBO, and we do not propose one. We will revise the manuscript to remove this incorrect terminology and instead clearly state that the ELBO derivation serves to motivate the regularization structure. Specifically, we plan to revise lines 770-774 as:
> > >
> > > "The ELBO derivation in Appendix B serves to derive the KL regularization terms implied by the GP prior under the Hilbert space parameterization; incorporating these into the OT framework yields the final objective in Equation (7)."
> > >
> > > The justification for using OT loss is twofold. First, the OT cost is computed in expectation over the variational distribution of the GP parameters (Eq. 7), so the transport cost remains internally consistent with the uncertainty and smoothness of the latent GP structure. Second, this objective can be understood through the lens of the Wasserstein autoencoder framework, which establishes that minimizing a transport cost in observation space combined with a latent regularization term constitutes a valid approach to generative modeling. We discuss this connection in Section 4.2.
> > >
> > > As the reviewer requested, we prepared a toy experiment where ground-truth cell pairings are available, allowing us to directly compare the effect of using a pointwise likelihood loss against an OT-based distributional loss.
> > >
> > > We construct a two-dimensional trajectory dataset with two lineages in which ground-truth cell pairings are available. We simulate 500 cells over 10 equally spaced time points ($\Delta t=1$). At $t=0$, all cells are drawn from a shared isotropic Gaussian $\mathbf{x} \sim \mathcal{N}(\mathbf{0}, \sigma^2 \mathbf{I})$, where $\sigma=0.1$. Each cell is assigned to one of two lineages with equal probability. Before the branching point $t=2$, every cell evolves under a common horizontal drift $v_x=0.5$ with no vertical component. After the branching point, lineage A acquires an additional upward drift $v_y=0.2$ while lineage B drifts downward by the same magnitude:
> > > $$\mathbf{x}_{t+1}=\mathbf{x}_t+\boldsymbol{\mu}_c^\top \Delta t+\boldsymbol{\varepsilon},$$
> > > where $\boldsymbol{\mu}_c=[v_x, v_y]^\top$ for lineage A and $\boldsymbol{\mu}_c=[v_x, -v_y]^\top$ for lineage B when $t\geq2$, and $\boldsymbol{\mu}_c=[v_x, 0]^\top$ otherwise.
> > >
> > > We compare three training regimes:
> > > - ELBO (paired): Standard ELBO training with Gaussian likelihood using ground-truth cell correspondences across time. We emphasize that a time-series scRNA-seq experiment cannot produce this type of "paired" data. Training on "paired" data is included here only for demonstration purposes.
> > > - ELBO (unpaired): The same ELBO objective, but with the ground-truth pairings of cells across time points broken. This reflects the characteristics of real temporal scRNA-seq data. For ELBO-based training, we must be able to evaluate the likelihood loss pointwise. Since the ground-truth pairing is not available, we need to randomly pair cells between consecutive time points at each training iteration, reflecting the absence of correspondence.
> > > - OT: Our OT-based objective, which does not require pairings by construction.
> > >
> > > The latent trajectory of each individual cell is learned by conditioning on the cell's ID, analogously to the cell type conditioning described in Section 4.4. In this setup, we expect ELBO (paired) to achieve the best performance, as it has access to individual cell trajectories, while the OT-based loss should perform competitively despite not using pairing information. ELBO (unpaired) serves as a control to illustrate the failure mode of the ELBO with a pointwise likelihood loss when correspondences are unavailable.
> > >
> > > We train models on the trajectory while holding intermediate time points (t=4, 6, and 8) out for evaluation. Below, we report the $W_2$ distance and a visualization of the reconstructed trajectories: https://ibb.co/TM86tb9n.
> > >
> > > ||4|6|8|
> > > -|-|-|-|
> > > ELBO (paired)|0.09|0.09|0.23|
> > > ELBO (unpaired)|0.47|0.86|1.26|
> > > OT|0.11|0.17|0.28|
> > >
> > > The OT-based objective achieves $W_2$ comparable to the paired ELBO training. When pairings are unavailable, the realistic setting for scRNA-seq data, the ELBO-based approach with broken pairings degrades noticeably; the model collapses onto the mean trajectory and fails to capture the spread, as is clearly visible in the middle row of the figure, yielding a substantially worse $W_2$. This toy experiment validates that the OT loss is an effective substitute in the unpaired setting, which is the only setting available for real scRNA-seq data.
> > >
> > > We will add the Sinkhorn algorithm reference to the manuscript. Thank you for the insightful discussion.
> > >
> > > Sinkhorn, R. (1964). A relationship between arbitrary positive matrices and doubly stochastic matrices.
> > >
> > > Sinkhorn, R., & Knopp, P. (1967). Concerning nonnegative matrices and doubly stochastic matrices.

---

### Decision · Program_Chairs · 2026-04-30

**Decision:**

Accept (regular)

**Comment:**

This paper proposes a decoder-only generative modeling framework for temporal single-cell RNA sequence (scRNA-seq) data that models population trends using a latent heteroscedastic Gaussian process. The framework links noisy scRNA-seq data to unobserved population trends via an optimal transport (OT) objective. Reviewers appreciated that the paper is well-written, and tackles an important applied problem. The reviewers also found the methodology to be original and technically sound. Initially, two concerns were raised about the soundness of including the OT objective and the encoder-free architecture, however these were assuaged by the authors' responses during the discussion period. In addition, some concerns about novelty were raised, but these were also assuaged, as the novelty was clarified as integrating known components, with that integration being well-motivated by genuine constraints of the domain. The camera-ready should clarify these points, but the paper is already in good shape.